# Finding, visualizing, and quantifying latent structure across diverse animal vocal repertoires

**Tim Sainburg**[1,2]*, **Marvin Thielk**[3], **Timothy Q. Gentner**[1,3,4,5]

**1** Department of Psychology, University of California, San Diego, La Jolla, CA, USA, **2** Center for Academic Research & Training in Anthropogeny, University of California, San Diego, La Jolla, CA, USA, **3** Neurosciences Graduate Program, University of California, San Diego, La Jolla, CA, USA, **4** Neurobiology Section, Division of Biological Sciences, University of California, San Diego, La Jolla, CA, USA, **5** Kavli Institute for Brain and Mind, University of California, San Diego, La Jolla, CA, USA

* tsainburg@ucsd.edu

**Data Availability Statement:** All of the vocalization datasets used in this study were acquired from external sources, most of them hosted publicly online (See Table 1). The data needed to reproduce

## Abstract

Animals produce vocalizations that range in complexity from a single repeated call to hundreds of unique vocal elements patterned in sequences unfolding over hours. Characterizing complex vocalizations can require considerable effort and a deep intuition about each species' vocal behavior. Even with a great deal of experience, human characterizations of animal communication can be affected by human perceptual biases. We present a set of computational methods for projecting animal vocalizations into low dimensional latent representational spaces that are directly learned from the spectrograms of vocal signals. We apply these methods to diverse datasets from over 20 species, including humans, bats, songbirds, mice, cetaceans, and nonhuman primates. Latent projections uncover complex features of data in visually intuitive and quantifiable ways, enabling high-powered comparative analyses of vocal acoustics. We introduce methods for analyzing vocalizations as both discrete sequences and as continuous latent variables. Each method can be used to disentangle complex spectro-temporal structure and observe long-timescale organization in communication.

## Author summary

Of the thousands of species that communicate vocally, the repertoires of only a tiny minority have been characterized or studied in detail. This is due, in large part, to traditional analysis methods that require a high level of expertise that is hard to develop and often species-specific. Here, we present a set of unsupervised methods to project animal vocalizations into latent feature spaces to quantitatively compare and develop visual intuitions about animal vocalizations. We demonstrate these methods across a series of analyses over 19 datasets of animal vocalizations from 29 different species, including songbirds, mice, monkeys, humans, and whales. We show how learned latent feature spaces untangle complex spectro-temporal structure, enable cross-species comparisons, and uncover

our results can be found on Zenodo (https://zenodo.org/record/3775893#.X3YdqZNKhTY).

**Funding:** Work supported by NSF (https://www.nsf.gov/) GRF 2017216247 and an Annette Merle-Smith Fellowship (https://carta.anthropogeny.org/support/fellowship) to T.S. and NIH (https://www.nih.gov/) DC0164081 and DC018055 to T.Q.G. The funders had no role in study design, data collection and analysis, decision to publish, or preparation of the manuscript.

**Competing interests:** The authors have declared that no competing interests exist.

high-level attributes of vocalizations such as stereotypy in vocal element clusters, population regiolects, coarticulation, and individual identity.

## Introduction

Vocal communication is a common social behavior among many species, in which acoustic signals are transmitted from sender to receiver to convey information such as identity, individual fitness, or the presence of danger. Across diverse fields, a set of shared research questions seeks to uncover the structure and mechanism of vocal communication: What information is carried within signals? How are signals produced and perceived? How does the communicative transmission of information affect fitness and reproductive success? Many methods are available to address these questions quantitatively, most of which are founded on underlying principles of abstraction and characterization of 'units' in the vocal time series [1]. For example, segmentation of birdsong into temporally discrete elements followed by clustering into discrete categories has played a crucial role in understanding syntactic structure in birdsong [1–9].

The characterization and abstraction of vocal communication signals remains both an art and a science. In a recent survey, Kershenbaum et. al. [1] outline four common steps used in many analyses to abstract and describe vocal sequences: (1) the collection of data, (2) segmentation of vocalizations into units, (3) characterization of sequences, and (4) identification of meaning. A number of heuristics guide these steps, but it is largely up to the experimenter to determine which heuristics to apply and how. This application typically requires expert-level knowledge, which in turn can be difficult and time-consuming to acquire, and often unique to the structure of each species' vocal repertoire. For instance, what constitutes a 'unit' of humpback whale song? Do these units generalize to other species? Should they? When such intuitions are available they should be considered, of course, but they are generally rare in comparison to the wide range of communication signals observed naturally. As a result, communication remains understudied in most of the thousands of vocally communicating species. Even in well-documented model species, characterizations of vocalizations are often influenced by human perceptual and cognitive biases [1, 10–12]. We explore a class of unsupervised, computational, machine learning techniques that avoid many of the foregoing limitations, and provide an alternative method to characterize vocal communication signals. Machine learning methods are designed to capture statistical patterns in complex datasets and have flourished in many domains [13, 14, 14–16, 16, 17]. These techniques are therefore well suited to quantitatively investigate complex statistical structure in vocal repertoires that otherwise rely upon expert intuitions. In this paper, we demonstrate the utility of unsupervised latent models, statistical models that learn latent (compressed) representations of complex data, in describing animal communication.

### Latent models of acoustic communication

Dimensionality reduction refers to the compression of high-dimensional data into a smaller number of dimensions, while retaining the structure and variance present in the original high-dimensional data. Each point in the high-dimensional input space can be projected into the lower-dimensional 'latent' feature space, and dimensions of the latent space can be thought of as features of the dataset. Animal vocalizations are good targets for dimensionality reduction. They appear naturally as sound pressure waveforms with rich, multi-dimensional temporal and spectral variations, but can generally be explained by lower-dimensional dynamics

[18–20]. Dimensionality reduction, therefore, offers a way to infer a smaller set of latent dimensions (or features) that can explain much of the variance in high-dimensional vocalizations.

The common practice of developing a set of basis-features on which vocalizations can be quantitatively compared *(also called Predefined Acoustic Features, or PAFs)* is a form of dimensionality reduction and comes standard in most animal vocalization analysis software (e.g. Luscinia [21], Sound Analysis Pro [22, 23], BioSound [24], Avisoft [25], and Raven [26]). Birdsong, for example, is often analyzed on the basis of features such as amplitude envelope, Weiner entropy, spectral continuity, pitch, duration, and frequency modulation [1, 22]. Grouping elements of animal vocalizations (e.g. syllables of birdsong, mouse ultrasonic vocalizations) into abstracted discrete categories is also a form of dimensionality reduction, where each category is a single orthogonal dimension. In machine learning parlance, the process of determining the relevant features, or dimensions, of a particular dataset, is called *feature engineering*. Engineered features are ideal for many analyses because they are human-interpretable in models that describe the relative contribution of those features as explanatory variables, for example explaining the contribution of the fundamental frequency of a *coo* call in predicting caller identity in macaques [27]. As with other human-centric heuristics, however, feature engineering has two caveats. First, the features selected by humans can be biased by human perceptual systems, which are not necessarily "tuned" for analyzing non-human communication signals [10, 28]. Second, feature engineering typically requires significant domain knowledge, which is time-consuming to acquire and difficult to generalize across species, impairing cross-species comparisons.

An attractive alternative to feature engineering is to project animal vocalizations into low-dimensional feature spaces that are determined directly from the structure of the data. Many methods for data-driven dimensionality reduction are available. PCA, for example, projects data onto a lower-dimensional surface that maximizes the variance of the projected data [1, 29], while multidimensional scaling (MDS) projects data onto a lower-dimensional surface that maximally preserves the pairwise distances between data points. Both PCA and MDS are capable of learning manifolds that are linear or near-linear transformations of the original high-dimensional data space [30].

More recently developed graph-based methods extend dimensionality reduction to infer latent manifolds as non-linear transformations of the original high-dimensional space using ideas from topology [30–32]. Like their linear predecessors, these non-linear dimensionality reduction algorithms also try to find a low-dimensional manifold that captures variation in the higher-dimensional input data, but the graph-based methods allow the manifold to be continuously deformed, by for example stretching, twisting, and/or shrinking, in high dimensional space. These algorithms work by building a topological representation of the data and then learning a low-dimensional embedding that preserves the structure of the topological representation (Fig 1). For example, while MDS learns a low-dimensional embedding that preserves the pairwise distance between points in Euclidean space, ISOMAP [30], one of the original topological non-linear dimensionality reduction algorithms, infers a graphical representation of the data and then performs MDS on the pairwise distances between points within the graph (geodesics) rather than in Euclidean space. These graph-based methods are often preferable to linear methods because they capture more of the local structure of the data, but these benefits do have a cost. Whereas the latent dimensions of PCA, for example, have a ready interpretation in terms of the variance in the data, the ISOMAP dimensions have no specific meaning beyond separability [31]. In addition, in practice, high-level (global) structure in the dataset, like the distances between clusters in low-dimensional embeddings, can be less meaningful in graph-based dimensionality reduction than in PCA or MDS, because current graph-based

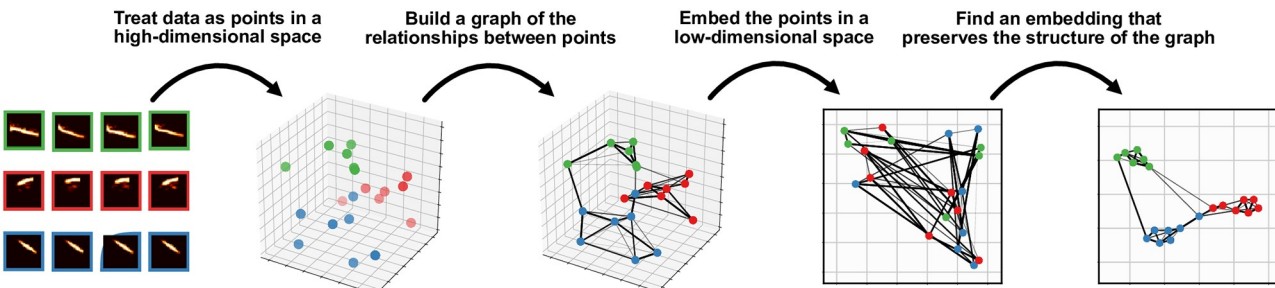

**Fig 1. Graph-based dimensionality reduction.** Current non-linear dimensionality reduction algorithms like TSNE, UMAP, and ISOMAP work by building a graph representing the relationships between high-dimensional data points, projecting those data points into a low-dimensional space, and then finds and embedding that retains the structure of the graph. This figure is for visualization, the spectrograms do not actually correspond to the points in the 3D space.

methods tend to local-notions of distance like nearest neighbors to construct a graphical representation [33].

The utility of non-linear dimensionality reduction techniques are just now coming to fruition in the study of animal communication, for example using t-distributed stochastic neighborhood embedding (t-SNE; [32]) to describe the development of zebra finch song [34], using Uniform Manifold Approximation and Projection (UMAP; [31]) to describe and infer categories in birdsong [3, 35], or using deep neural networks to synthesize naturalistic acoustic stimuli [36, 37]. Developments in non-linear representation learning have helped fuel the most recent advancements in machine learning, untangling statistical relationships in ways that provide more explanatory power over data than traditional linear techniques [13, 14]. These advances have proven important for understanding data in diverse fields including the life sciences (e.g. [3, 16, 34, 35, 38, 39]), in part due to their utility in rapidly extracting complex features from increasingly large and high-dimensional datasets.

In this paper, we describe a class of nonlinear latent models that learn complex feature-spaces of vocalizations, requiring few *a priori* assumptions about the features that best describe a species' vocalizations. We show that these methods reveal informative, low-dimensional, feature-spaces that enable the formulation and testing of hypotheses about animal communication. We apply our method to diverse datasets consisting of over 20 species (S1 Table), including humans, bats, songbirds, mice, cetaceans, and nonhuman primates. We introduce methods for treating vocalizations both as sequences of temporally discrete elements such as syllables, as is traditional in studying animal communication [1], as well as temporally continuous trajectories, as is becoming increasingly common in representing neural sequences [40]. Using both methods, we show that latent projections produce visually-intuitive and quantifiable representations that capture complex acoustic features. We show comparatively that the spectrotemporal characteristics of vocal units vary from species to species in how distributionally discrete they are and discuss the relative utility of different ways to represent different communicative signals.

## Results

### Dimensionality reduction

The current state-of-the-art graph-based manifold learning algorithms are t-SNE [32] and UMAP [31]. Like ISOMAP, t-SNE and UMAP first build a topological (graphical) representation of the data, and then project that graph into a lower-dimensional embedding, preserving as much of the topological structure of the graph as possible. Both embedding methods are

unsupervised, meaning they do not require labeled data. To visually compare the graph-based dimensionality reduction algorithms UMAP and t-SNE to the more classical linear methods PCA and MDS, we projected spectrograms of a dataset of Egyptian fruit bat infant isolation calls from 12 individuals into 2-dimensional PCA, MDS, t-SNE, and UMAP (Fig 2). Broadly,

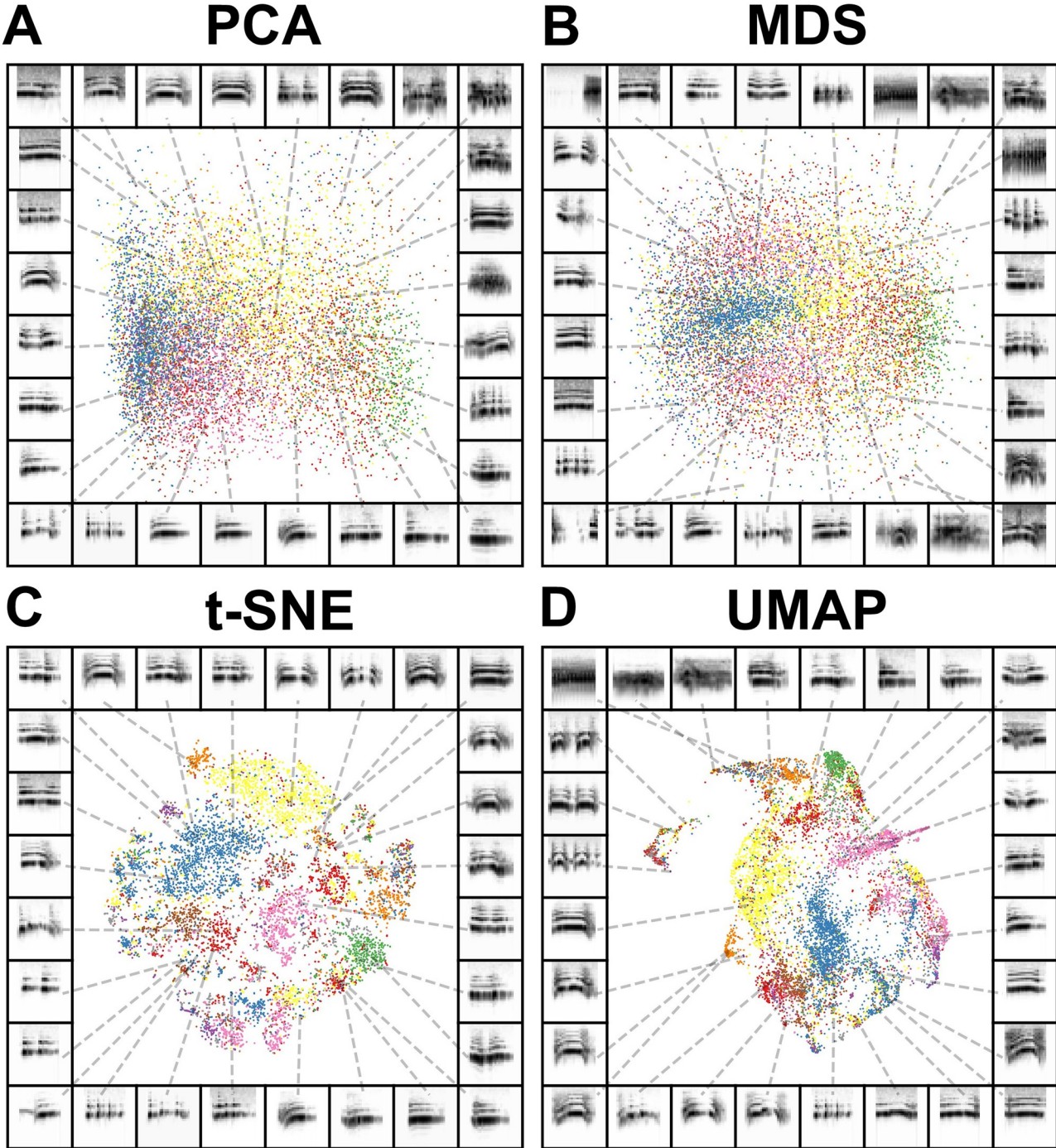

**Fig 2. Comparison between dimensionality reduction and manifold learning algorithms.** Isolation calls from 12 juvenile Egyptian fruit bats, where spectrograms of vocalizations are projected into two dimensions in (A) PCA, (B) MDS, (C) t-SNE, and (D) UMAP. In each panel, each point in the scatterplot corresponds to a single isolation call. The color of each point corresponds to the ID of the caller. The frame of each panel is a spectrogram of an example syllable, pointing to where that syllable lies in the projection.

we can see that PCA and MDS projections are more diffuse (Fig 2A and 2B), while t-SNE and UMAP capture much more of the local similarity structure across the dataset, tightly packing together calls from the same individuals (Fig 2C and 2D).

Throughout this manuscript, we chose to use UMAP over t-SNE because UMAP has been shown to preserve more global structure, decrease computation time, and effectively produce more meaningful data representations across a number of datasets within the natural sciences (e.g. [3, 16, 31, 35]).

Both t-SNE and UMAP are underlied by functionally similar steps: (1) construct a probabilistically weighted graph and (2) embed the graph in a low-dimensional embedding space (see Fig 1). To build a probabilistically weighted graph, UMAP and t-SNE first build a nearest-neighbor graph of the high-dimensional data using some distance metric (e.g. the Euclidean distance between spectrograms). They then compute a probability distribution over the edges of that graph (pairs of nearest neighbors), assigning higher weights to closer pairs, and lower weights to more distant pairs. Embedding that graph in lower-dimensional space is then simply a graph-layout problem. An embedding is first initialized (e.g. using PCA or a spectral embedding of the graph). UMAP and t-SNE then compute the probabilities over the relationships between projections in the embedding space, again where closer pairs of elements are assigned a higher probability and more distant pairs are assigned a lower probability. Using gradient-descent, the embeddings are then optimized to minimize the difference between the probability distribution computed from the nearest-neighbor graph and the probability distribution in the embedding space.

UMAP and t-SNE differ in how these graphs are constructed and how embeddings are optimized. UMAP, in particular, assumes that the high-dimensional space in which the data lives is warped, such that data are uniformly distributed on a non-linear manifold in the original dataspace. UMAP's construction of the graphical representation of the data uses concepts from topology, so that the edges of the graph (the connections between data points) are probabilistically weighted by distance on the uniform manifold. The embeddings are then found by minimizing the cross-entropy between the graph and a probability distribution defined over the relationships between embeddings. In other words, an embedding is learned that tries to preserve as much of the topological structure of the original graph as possible.

UMAP has several parameters for constructing its graph and embedding it in a low-dimensional space. The four primary UMAP parameters are `n_neighbors` which determines how many neighbors (nearby data points) are used in constructing the nearest-neighbor graph, `min_dist` which determines how spread apart connected embedding are allowed to be, `n_components` which is the dimensionality of the embedding space, and `metric` which defines the distance metric (e.g. Euclidean, cosine) that is used to define distances between points in the high-dimensional dataspace. We use the default parameters for each, except when otherwise noted.

## Choosing features to represent vocalizations

Choosing the best features to represent vocal data is difficult without significant domain knowledge. In some species, the features underlying behaviorally-relevant variability in vocalizations are well documented and understood. When such information about a species' vocal repertoire is known, those features can and should be used to make comparisons between vocalizations within species. When analyzing vocalizations across species or within species whose vocal repertoires are less well understood, choosing features to represent vocalizations is more difficult: features that capture only a subset of the true behaviorally relevant variance can bias downstream analyses in unforeseen ways.

Two methods for choosing feature-sets are commonly used by experimenters when the features underlying vocal data are unknown: (1) extract common descriptive statistics of vocalizations, sometimes called Predefined Acoustical Features (PAFs; e.g. mean fundamental frequency, syllable length, spectral entropy) and make comparisons on the basis of PAFs, or (2) make comparisons based upon time-frequency representations of the data (i.e. spectrograms) where the magnitude of each time-frequency component in the spectrogram is treated as an independent feature (or dimension) of the vocalization.

To compare and visualize the structure captured by both PAF and spectrogram representations of vocalizations, we used a subset of the 20 most frequent syllable-types from a dataset of Cassin's vireo song recorded in the Sierra Nevada Mountains [7, 41]. We computed both spectrographic representations of syllables as well as a set of 18 temporal, spectral, and fundamental characteristics (S2 Table) over each syllable using the BioSound python package [24]. We then projected both the spectral representation as well as the PAFs into 2D UMAP feature spaces (Fig 3A and 3B). To quantify the difference in how well clustered the different data representations are, we compare the silhouette score (Eq 4; [42]) of each representation. The silhouette score is a measure of how well a dataset is clustered relative to a set of known category labels (e.g. syllable label, species identity). The silhouette score is the mean silhouette coefficient across all of the samples in a dataset, where the silhouette coefficient measures how distant each point is to points in its own category, relative to its distance from the nearest point in another category. It is therefore taken as a measure of how well clustered together elements are that belong to the same category. Silhouette scores range from -1 to 1, with 1 being more clustered.

Overall, the UMAP projections significantly increase the clusterability of syllables in the Cassin's vireo dataset. The UMAP representations of both the PAF and the spectrogram data (Fig 3A and 3B) are more clustered than either PAFs or spectrograms alone. The silhouette score of PAFs (0.054) is significantly lower than that for the UMAP projections of PAFs (0.092; $H(2) = 632$; $p < 10^{-10}$; Fig 3A), and the silhouette score of spectrograms (0.252) is significantly lower than that of the UMAP projections of spectrograms (0.772; $H(2) = 37868$; $p < 10^{-10}$; Fig 3B). In addition, comparing between features, the UMAP projections of spectrograms yields more clearly discriminable clusters than UMAP projections of the PAFs ($H(2) = 37868$; $p < 10^{-10}$). All the silhouette scores are significantly better than chance (for each, $H(2) > 500$; $p < 10^{-10}$; see Methods). Thus, for this dataset, UMAP projections yield highly clusterable representations of the data points, and UMAP projections of spectrograms are more clustered than UMAP projections of PAFs. One should not infer from this, however, that spectrographic representations necessarily capture more structure than PAFs in all cases. For zebra finch vocalizations, PAFs provide more information about vocalization types than spectrograms [28], and in other datasets, smaller basis sets of acoustic features can account for nearly all the dynamics of a vocal element (e.g. [43]). Even when spectrographic representations are more clearly clusterable than PAFs, knowing how explicit features of data (e.g. fundamental frequency) are related to variability can be more useful than being able to capture variability in the feature space without an intuitive understanding of what those features represent. These different representations may capture different components of the signals. To highlight this, we show how two PAFs (Mean Fundamental Frequency and Pitch Saliency) vary within spectrographic UMAP clusters (Fig 3C and 3D), by overlaying the color-coded PAFs onto the UMAP projections of the spectrographic representations from Fig 3B). The relationships between PAFs and UMAP spectrogram projections exemplifies the variability of different PAFs within clusters, as well as the non-linear relationships learned by UMAP projections. Additional PAFs overlaid on UMAP projections are shown in S1 Fig.

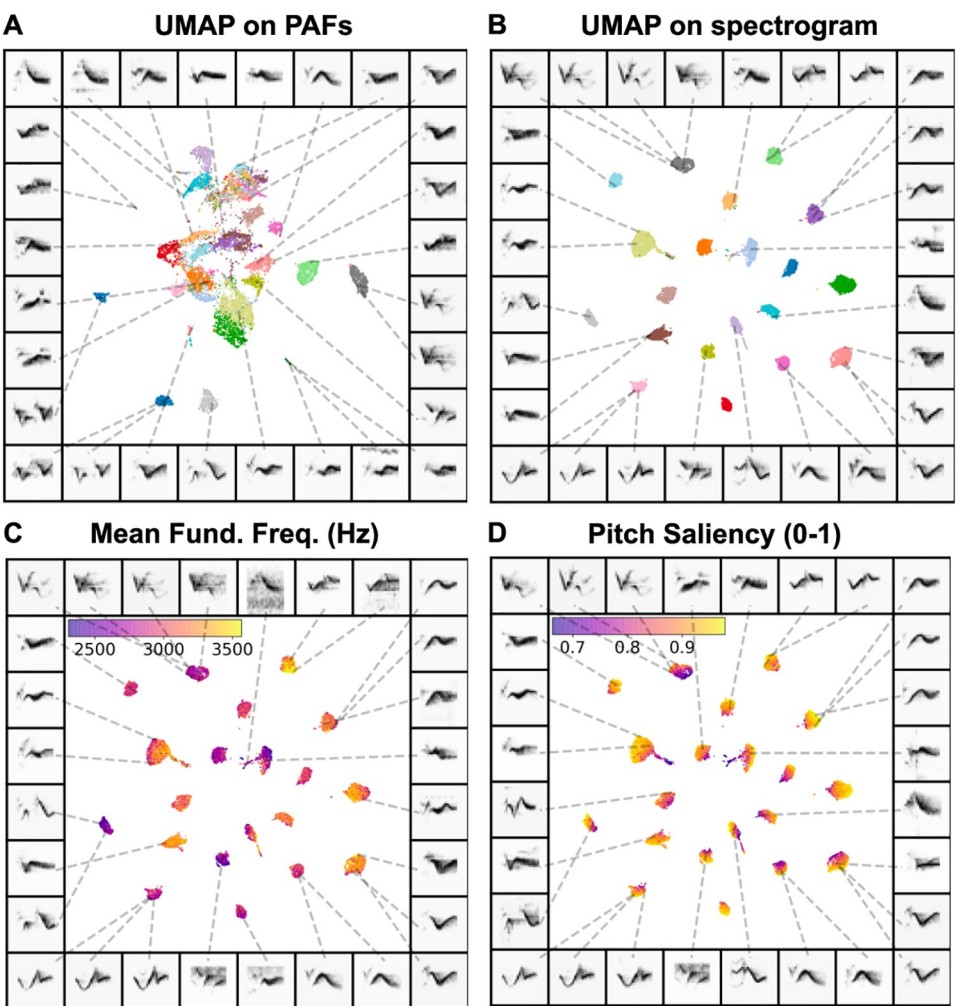

**Fig 3. Comparison between dimensionality reduction on spectrograms versus computed features of syllables.** Each plot shows 20 syllables of Cassin's vireo song. (A) UMAP projections of 18 features (see S2 Table) of syllables generated using BioSound. (B) UMAP applied to spectrograms of syllables. (C) UMAP of spectrograms where color is the syllable's average fundamental frequency (D) The same as (C) where pitch saliency of each syllable, which corresponds to the relative size of the first auto-correlation peak represents color.

## Discrete latent projections of animal vocalizations

To explore the broad utility of latent models in capturing features of vocal repertoires, we analyzed nineteen datasets consisting of 400 hours of vocalizations and over 3,000,000 discrete vocal units from 29 unique species (S1 Table). Each vocalization dataset was temporally segmented into discrete units (e.g. syllables, notes), either based upon segmentation boundaries provided by the dataset (where available), or using a novel dynamic-thresholding segmentation algorithm that segments syllables of vocalizations between detected pauses in the vocal stream (See Segmentation). Each dataset was chosen because it contains large repertoires of vocalizations from relatively acoustically isolated individuals that can be cleanly separated into temporally-discrete vocal units. With each temporally discrete vocal unit we computed a spectrographic representation (S2 Fig; See Spectrogramming). We then projected the spectrograms into latent feature spaces using UMAP. From these latent feature spaces, we analyzed datasets

for classic vocal features of animal communication signals, speech features, stereotypy/cluster-ability, and sequential organization.

**Vocal features.** Latent non-linear projections often untangle complex features of data in human-interpretable ways. For example, the latent spaces of some neural networks linearize the presence of a beard in an image of a face without being trained on beards in any explicit way [15, 44]. Complex features of vocalizations are similarly captured in intuitive ways in latent projections [3, 35–37]. Depending on the organization of the dataset projected into a latent space, these features can extend over biologically or psychologically relevant scales. Accordingly, we used our latent models to look at spectro-temporal structure within the vocal repertoires of individual's, and across individuals, populations, and phylogeny. These latent projections capture a range of complex features, including individual identity, species identity, linguistic features, syllabic categories, and geographical variability. We discuss each of these complex features in more detail below.

**Individual identity.** Many species produce caller-specific vocalizations that facilitate the identification of individuals when other sensory cues, such as sight, are not available. The features of vocalizations facilitating individual identification vary between species. We projected identity call datasets (i.e., sets of calls thought to carry individual identity information) from four different species into UMAP latent spaces (one per species) to observe whether individual identity falls out naturally within the latent space.

We looked at four datasets where both caller and call-type are available. Caller identity is evident in latent projections of all four datasets (Fig 4). The first dataset is comprised of macaque coo calls, where identity information is thought to be distributed across multiple features including fundamental frequency, duration, and Weiner entropy [27]. Indeed, the latent projection of coo calls clustered tightly by individual identity (silhouette score = 0.378; Fig 4A). The same is true for zebra finch distance calls [28] (silhouette score = 0.615; Fig 4B). Egyptian fruit bat pup isolation calls, which in other bat species are discriminable by adult females [45, 45, 46] clearly show regions of UMAP space densely occupied by single individual's vocalizations, but no clear clusters (silhouette score = -0.078; Fig 4C). In the marmoset phee call dataset [47] it is perhaps interesting that given the range of potential features thought to carry individual identity [27], phee calls appear to lie along a single continuum where each individual's calls occupy overlapping regions of the continuum (silhouette score = -0.062; Fig 4D). The silhouette score for each species was well above chance (H(2) > 20, p < $10^{-5}$). These patterns predict that some calls, such as macaque *coo* calls, would be more easily discriminable by conspecifics than other calls, such as marmoset *phee* calls.

The latent projections of these datasets demonstrate that individual identity can be obtained from all these vocalizations. Importantly, this information is available without *a priori* knowledge of specific spectro-temporal features, which is likely also the case for the animals attempting to use it. Because no caller identity information is used in learning the latent projections, the emergence of this information indicates that the similarity of within-caller vocalizations contains enough statistical power to overcome variability between callers. This within-caller structure likely facilitates conspecific learning of individual identity without *a priori* expectations for the distribution of relevant features [48], in the same way that developing sensory systems adapt to natural environmental statistics [49].

**Cross species comparisons.** Classical comparative studies of vocalizations across species rely on experience with multiple species' vocal repertoires. This constrains comparisons to those species whose vocalizations are understood in similar feature spaces, or forces the choice of common feature spaces that may obscure relevant variation differently in different species. Because latent models learn arbitrary complex features of datasets, they can yield less biased comparisons between vocal repertoires where the relevant axes are unknown, and where the

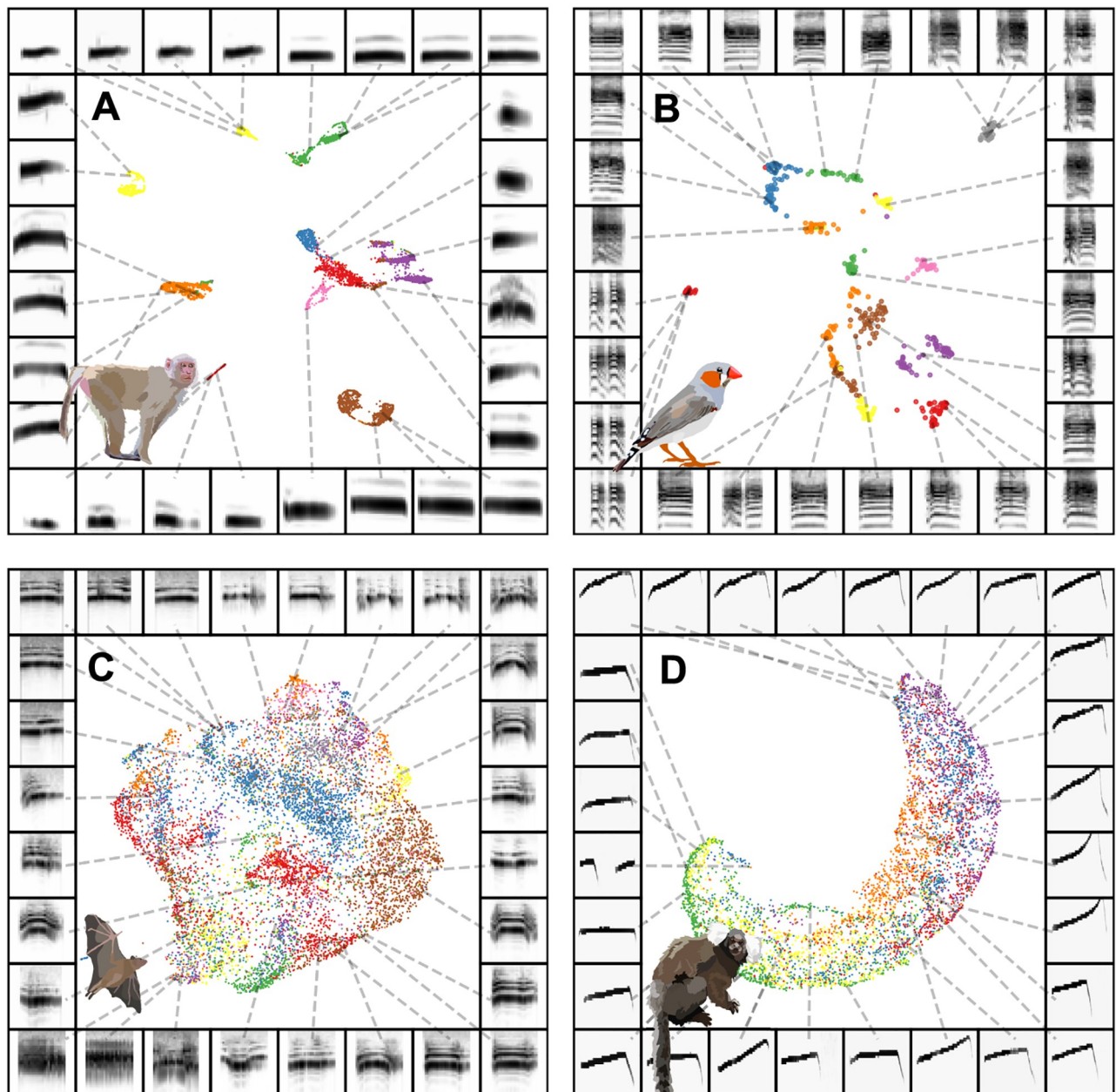

**Fig 4. Individual identity is captured in projections for some datasets.** Each plot shows vocal elements discretized, spectrogrammed, and then embedded into a 2D UMAP space, where each point in the scatterplot represents a single element (e.g. syllable of birdsong). Scatterplots are colored by individual identity. The borders around each plot are example spectrograms pointing toward different regions of the scatterplot. (A) Rhesus macaque coo calls. (B) Zebra finch distance calls. (C) Fruit bat infant isolation calls. (D) Marmoset phee calls.

surface structures are either very different, for example canary and starling song, or very similar, like the echolocation clicks of two closely related beaked whales.

To explore how well latent projections capture vocal repertoire variation across species, we projected a dataset containing monosyllabic vocalizations [50] from eleven different species of North American birds into UMAP latent space (silhouette score = 0.377), well above chance ($H(2) = 1396$, $p < 10^{-10}$). Similar "calls", like those from the American crow *caw* and great blue heron *roh* are closer together in latent space, while more distinct vocalizations, like chipping

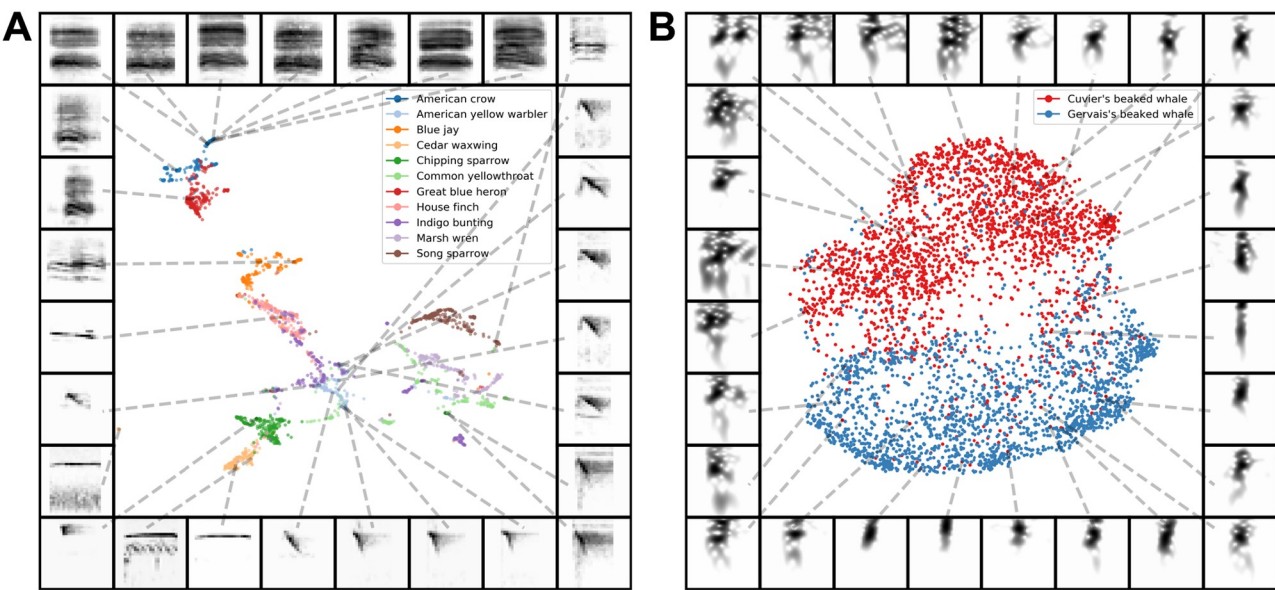

**Fig 5. Comparing species with latent projections.** (A) Calls from eleven species of North American birds are projected into the same UMAP latent space. (B) Cuvier's and Gervais's beaked whale echolocation clicks are projected into UMAP latent space and fall into two discrete clusters.

sparrow notes, are further apart (Fig 5A). Latent projections like this have the potential power to enable comparisons across broad phylogenies without requiring decisions about which acoustic features to compare.

At the other extreme is the common challenge in bioacoustics research to differentiate between species with very similar vocal repertoires. For example, Cuvier's and Gervais' beaked whales, two sympatric species recorded in the Gulf of Mexico, have echolocation clicks with highly overlapping power spectra that are generally differentiated using supervised learning approaches (c.f. [51, 52]). We projected a dataset containing Cuvier's and Gervais' beaked whale echolocation clicks into UMAP latent space. Species-identity again falls out nicely, with clicks assorting into distinct clusters that correspond to species (Fig 5B). The silhouette score of UMAP on the spectrogram (shown in Fig 5B) was 0.401, higher than the silhouette score of UMAP on the power spectra (0.171; H(2) = 2411; p < $10^{-10}$) which is in turn higher than the silhouette score of the power spectra alone (0.066; H(2) = 769; p < $10^{-10}$). Each silhouette score is also well above chance (H(2) > 500; p < $10^{-10}$). The utility of an approach such as UMAP to clustering echolocation clicks is perhaps unsurprising; recent work [52] has shown that graph-based methods are successful for representing and clustering echolocation clicks of a larger dataset of cetacean echolocation clicks.

**Population geography.** Some vocal learning species produce different vocal repertoires (regiolects) across populations occupying different geographic regions. Differences in regiolects between populations are borne out in the categorical perception of notes [53–55], much the same as cross-linguistic differences in the categorical perception of phonemes in human speech [56]. To compare vocalizations across geographical populations in the swamp sparrow, which produces regionally distinct trill-like songs [21], we projected individual notes into a UMAP latent space. Although the macro-structure of clusters suggest common note-types across multiple populations, most of the larger clusters show multiple clear sub-regions that are tied to vocal differences between geographical populations (Fig 6). We further explore how these projections of notes relate to vocal clusters in traditional feature spaces later in the manuscript.

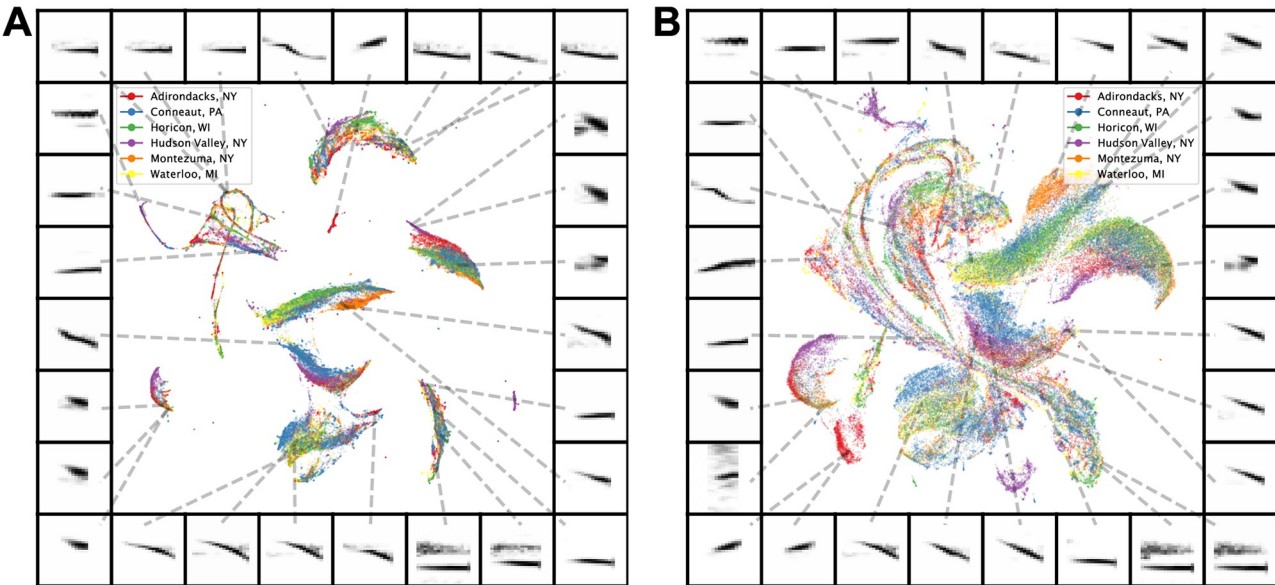

**Fig 6. Comparing notes of swamp sparrow song across different geographic populations.** (A) Notes of swamp sparrow song from six different geographical populations projected into a 2D UMAP feature space. (B) The same dataset from (A) projected into a 2D UMAP feature space where the parameter `min_dist` is set at 0.25 to visualize more spread in the projections.

**Phonological features.** The sound segments that make up spoken human language can be described by distinctive phonological features that are grouped according to articulation place and manner, glottal state, and vowel space. A natural way to look more closely at variation in phoneme production is to look at variation between phonemes that comprise the same phonological features. As an example, we projected sets of consonants that shared individual phonological features into UMAP latent space (Fig 7, S3 Fig). In most cases, individual phonemes tended to project to distinct regions of latent space based upon phonetic category, and consistent with their perceptual categorization. At the same time, we note that latent projections vary smoothly from one category to the next, rather than falling into discrete clusters. This provides a framework that could be used in future work to characterize the distributional properties of speech sounds in an unbiased manner. Likewise, it would be interesting to contrast projections of phonemes from multiple languages, in a similar manner as the swamp sparrow (Fig 6), to visualize and characterize variation in phonetic categories across languages [56].

**Variation in discrete distributions and stereotypy.** In species as phylogenetically diverse as songbirds and rock hyraxes, analyzing the sequential organization of communication relies upon similar methods of segmentation and categorization of discrete vocal elements [1]. In species such as the Bengalese finch, where syllables are highly stereotyped, clustering syllables into discrete categories is a natural way to abstract song. The utility of clustering song elements in other species, however, is more contentious because discrete category boundaries are not as easily discerned [10, 11, 35, 57].

To compare broad structural characteristics across a wide sampling of species, we projected vocalizations from 14 datasets of different species vocalizations, ranging across songbirds, cetaceans, primates, and rodents into UMAP space (Fig 8). To do so, we sampled from a diverse range of datasets, each of which was recorded from a different species in a different setting (S1 Table). Some datasets were recorded from single isolated individuals in a sound isolated chamber in a laboratory setting, while others were recorded from large numbers of freely behaving individuals in the wild. In addition, the units of vocalization from each dataset are variable.

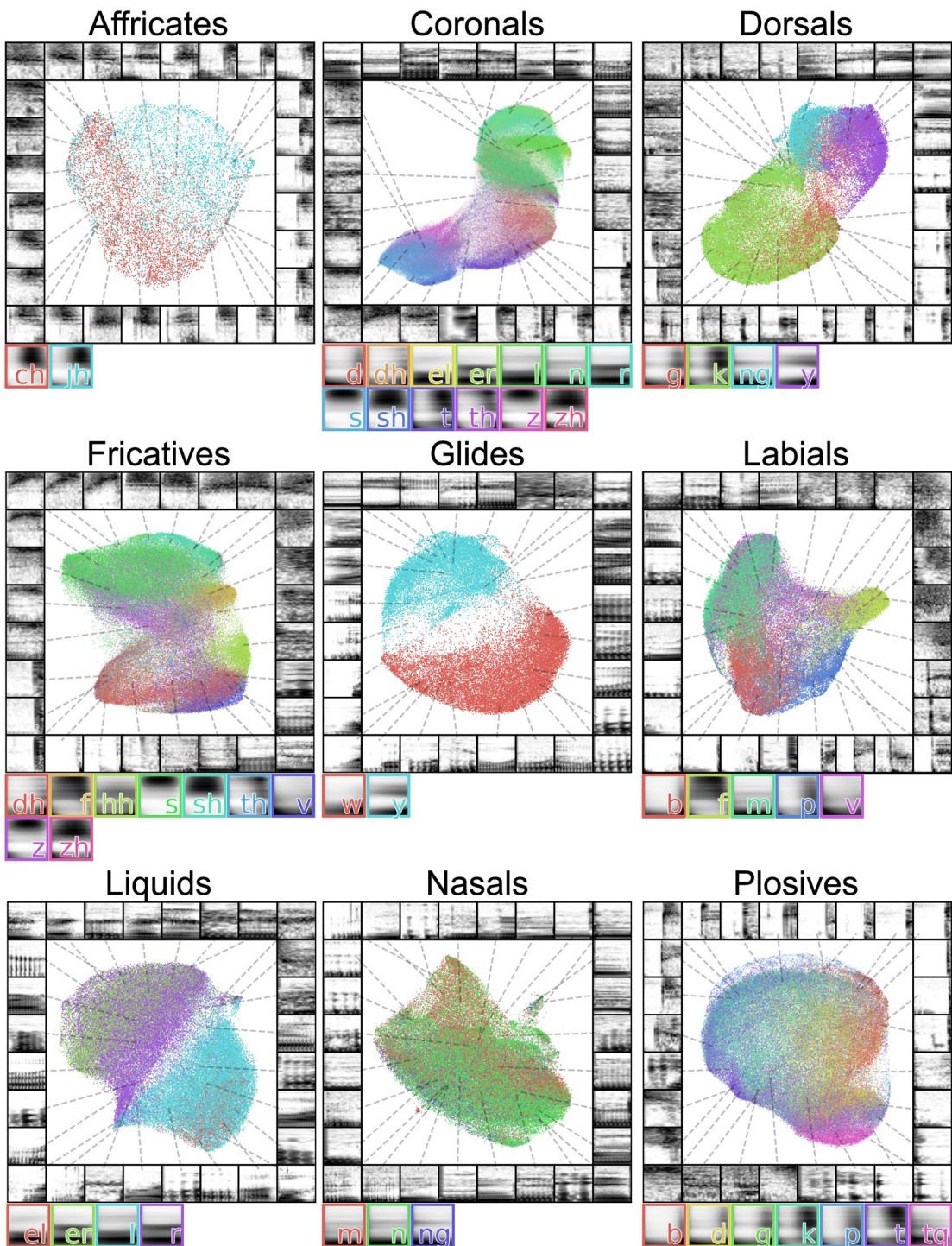

**Fig 7. Latent projections of consonants.** Each plot shows a different set of consonants grouped by phonetic features. The average spectrogram for each consonant is shown to the right of each plot.

We used the smallest units of each vocalization that could be easily segmented, for example, syllables, notes, and phonemes. Thus, this comparison across species is not well-controlled. Still, such a dataset enabling a broad comparison in a well-controlled manner does not exist. Latent projections of such diverse recordings, while limited in a number of ways, have the

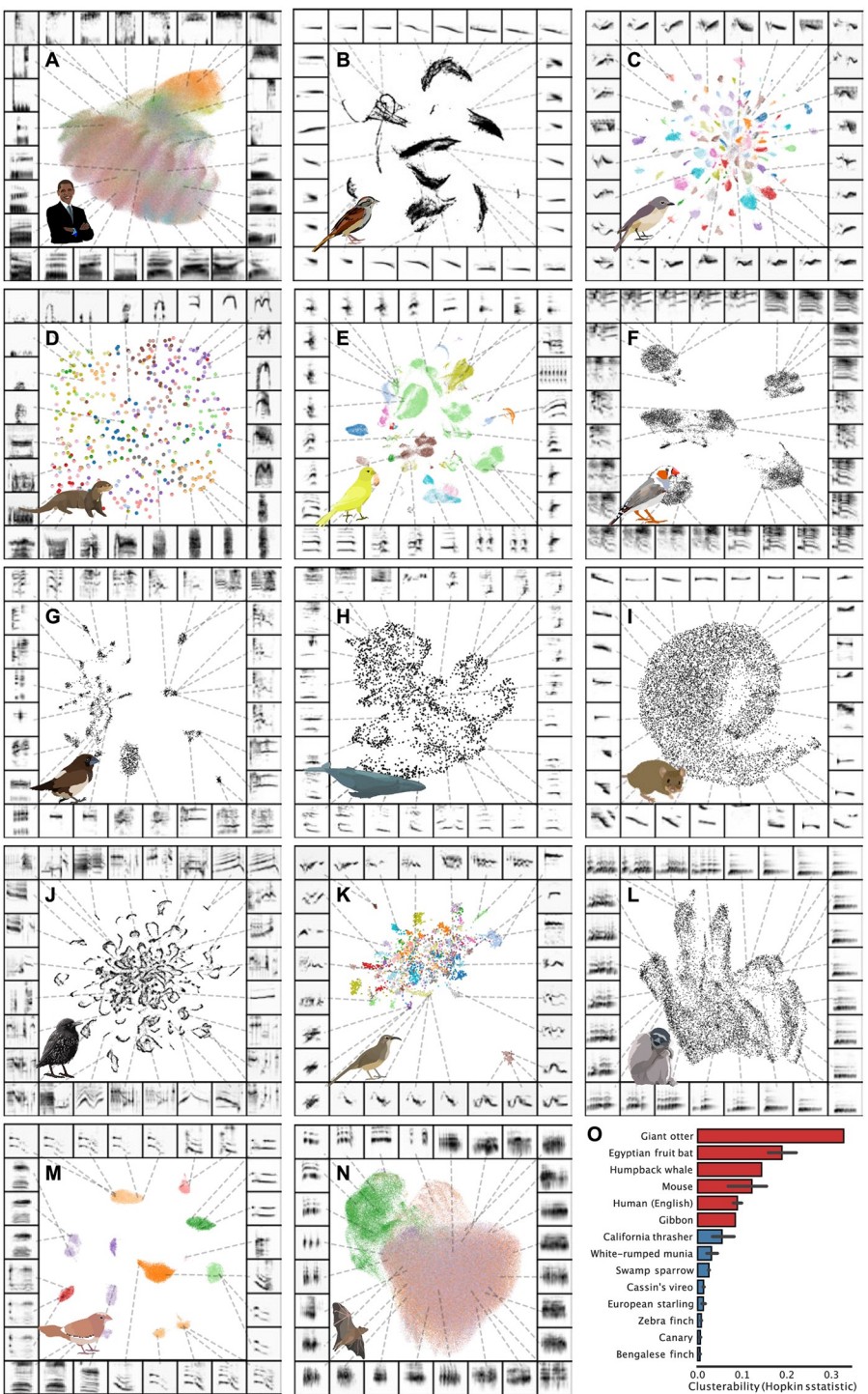

**Fig 8. UMAP projections of vocal repertoires across diverse species.** Each plot shows vocal elements segmented, spectrogrammed, and then embedded into a 2D UMAP space, where each point in the scatterplot represents a single element (e.g. syllable of birdsong). Scatterplots are colored by element categories over individual vocalizations as defined by the authors of each dataset, where available. Projections are shown for single individuals in datasets where vocal repertoires were visually observed to be distinct across individuals and a large dataset was available for single individuals (E, F, G, H, I, J, M). Projections are shown across individuals for the remainder of panels. (A) Human phonemes. (B) Swamp sparrow notes. (C) Cassin's vireo syllables. (D) Giant otter calls. (E) Canary syllables. (F) Zebra finch sub-motif syllables. (G) White-rumped munia syllables. (H) Humpback whale syllables. (I) Mouse USVs. (J) European starling syllables. (K) California thrasher syllables. (L) Gibbon syllables. (M) Bengalese finch syllables. (N)

Egyptian fruit bat calls (color is context). (O) Clusterability (Hopkin's metric) for each dataset. Lower is more clusterable. Hopkin's metric is computed over UMAP projected vocalizations for each species. Error bars show the 95% confidence interval across individuals. The Hopkin's metric for gibbon vocalizations and giant otter voalizations are shown across individuals, because no individual identity information was available. Color represents species category (red: mammal, blue: songbird).

potential to provide a glimpse into broad structure into vocal repertoires, yielding novel insights into broad trends in animal communication. For each dataset, we computed spectrograms of isolated elements, and projected those spectrograms into UMAP space (Fig 8). Where putative element labels are available, we plot them in color over each dataset.

Visually inspecting the latent projections of vocalizations reveals appreciable variability in how the repertoires of different species cluster in latent space. For example, mouse USVs appear as a single cluster (Fig 8I), while zebra finch syllables appear as multiple discrete clusters (Fig 8M and 8F), and gibbon song sits somewhere in between (Fig 8L). This suggests that the spectro-temporal acoustic diversity of vocal repertoires fall along a continuum ranging from unclustered and uni-modal to highly clustered.

We quantified this effect using a linear mixed-effects model comparing the Hopkin's statistic across UMAP projections of vocalizations from single individuals (n = 289), controlling for the number of vocalizations produced by each individual as well as random variability in clusterability at the level of species. We included each of the species in Fig 8 except giant otter and gibbon vocalizations, as individual identity was not available for those datasets. We find that songbird vocalizations are significantly more clustered than mammalian vocalizations ($\chi^2(1) =$ 20, p $< 10^{-5}$; See Methods).

The stereotypy of songbird (and other avian) vocal elements is well documented [58, 59] and at least in zebra finches is related to the high temporal precision in the singing-related neural activity of vocal-motor brain regions [60–62]. The observed differences in stereotypy between songbirds and mammals should be interpreted with consideration of the broad variability underlying the datasets, however.

**Clustering vocal element categories.**   UMAP projections of birdsongs largely fall more neatly into discriminable clusters (Fig 8). If clusters in latent space are highly similar to experimenter-labeled element categories, unsupervised latent clustering could provide an automated and less time-intensive alternative to hand-labeling elements of vocalizations. To examine this, we compared how well clusters in latent space correspond to experimenter-labeled categories in three human-labeled datasets: two separate Bengalese finch datasets [63, 64], and one Cassin's vireo dataset [7]. We compared four different labeling techniques: a hierarchical density-based clustering algorithm (HDBSCAN; Fig 9; [65, 66]) applied to UMAP projections of

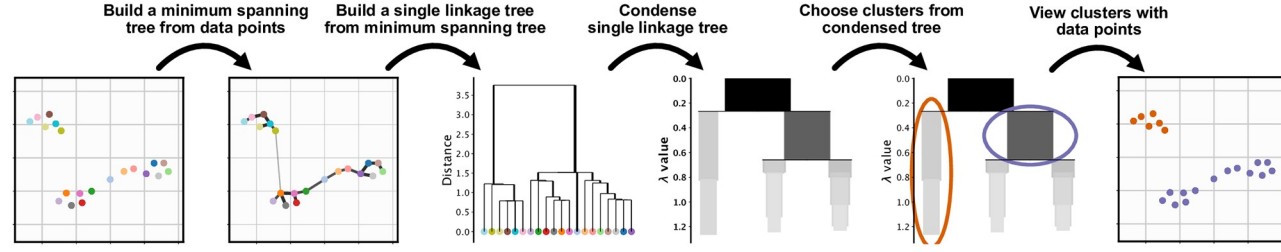

**Fig 9. HDBSCAN density-based clustering.** Clusters are found by generating a graphical representation of data, and then clustering on the graph. The data shown in this figure are from the latent projections from Fig 1. Notably, the three clusters in Fig 1. are clustered into only two clusters using HDBSCAN, exhibiting a potential shortcoming of the HDBSCAN algorithm. The grey colormap in the condensed trees represent the number of points in the branch of the tree. Λ is a value used to compute the persistence of clusters in the condensed trees.

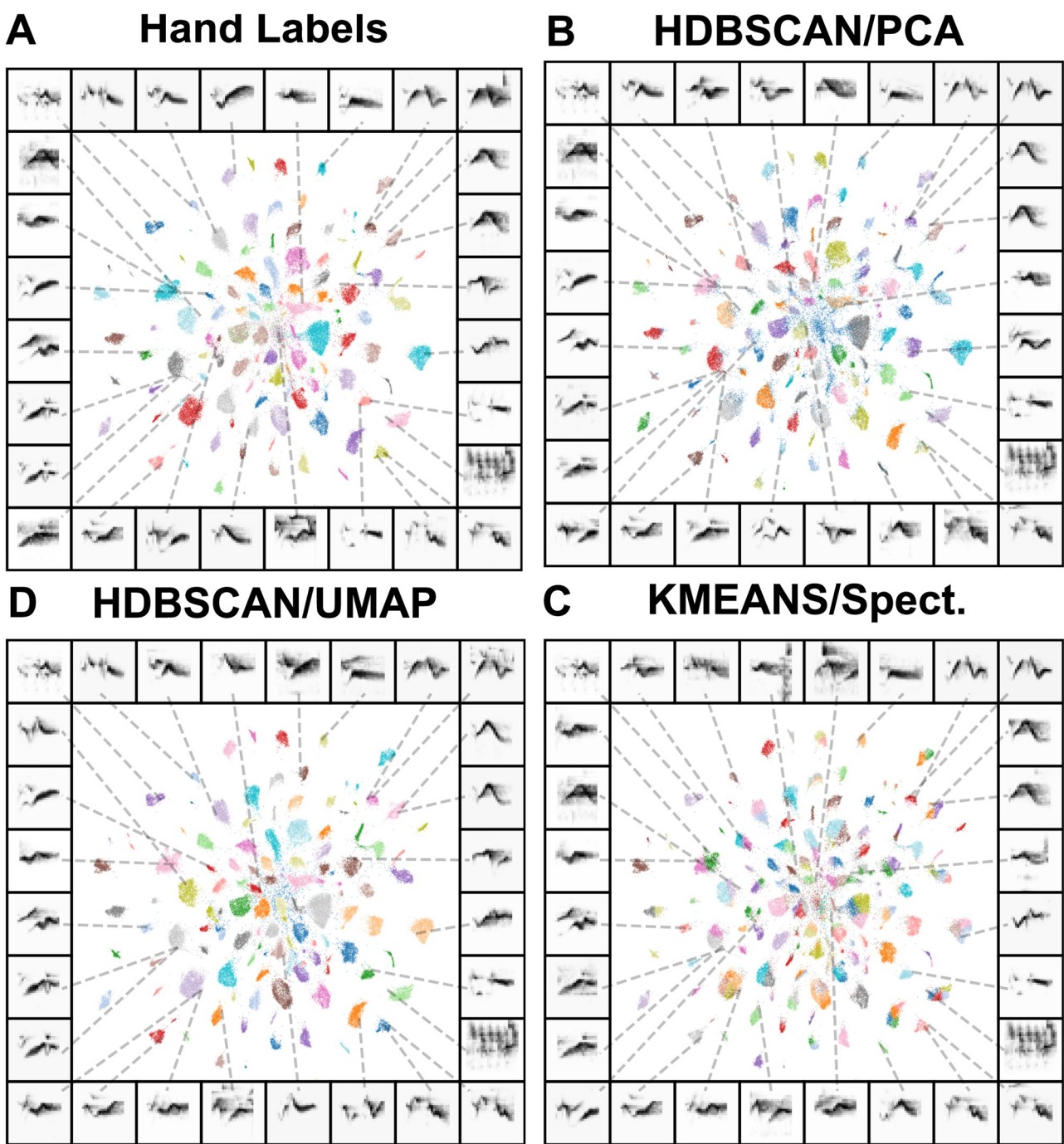

**Fig 10. Clustered UMAP projections of Cassin's vireo syllable spectrograms.** Panels (A-D) show the same scatterplot, where each point corresponds to a single syllable spectrogram projected into two UMAP dimensions. Points are colored by their hand-labeled categories (A), which generally fall into discrete clusters in UMAP space. Remaining panels show the same data colored according to cluster labels produced by (B) HDBSCAN over PCA projections (100 dimensions), (C) HDBSCAN on UMAP projections, and (D) k-means directly on syllable spectrograms.

spectrograms, HDBSCAN applied to PCA projections of spectrograms (HDBSCAN is applied to 100-dimensional PCA projections rather than spectrograms directly because HDBSCAN does not perform well in high-dimensional spaces [66]), k-means [67] clustering applied over UMAP, and k-means clustering applied over spectrograms (Fig 10; Table 1).

**Table 1. Cluster similarity to hand labels for two Bengalese finch and one Cassin's vireo dataset.** Four clustering methods were used: (1) KMeans on spectrograms (2) KMeans on UMAP projections (3) HDBSCAN on first 100 principal components of spectrograms (4) HDBSCAN clustering of UMAP projections. With KMeans 'K' was set to the correct number of clusters to make it more competitive with HDBSCAN clustering. Standard deviation across individual birds is shown for the finch datasets. Best performing method for each metric is bolded.

| | Homogeneity | Completeness | V-measure |
|---|---|---|---|
| **B. Finch (Koumura)** | | | |
| KMeans | 0.911±0.044 | 0.85±0.064 | 0.879±0.051 |
| KMeans/UMAP | 0.842±0.116 | 0.796±0.145 | 0.817±0.132 |
| HDBSCAN/PCA | 0.968±0.036 | **0.86±0.14** | **0.902±0.086** |
| HDBSCAN/UMAP | **0.99±0.006** | 0.74±0.122 | 0.841±0.088 |
| **B. Finch (Nicholson)** | | | |
| KMeans | 0.954±0.024 | 0.707±0.101 | 0.809±0.074 |
| KMeans/UMAP | **0.967±0.018** | 0.688±0.098 | 0.801±0.072 |
| HDBSCAN/PCA | 0.901±0.067 | 0.837±0.027 | 0.866±0.034 |
| HDBSCAN/UMAP | 0.963±0.022 | **0.855±0.076** | **0.903±0.042** |
| **Cassin's vireo** | | | |
| KMeans | 0.894 | 0.808 | 0.849 |
| KMeans/UMAP | 0.928 | 0.829 | 0.875 |
| HDBSCAN/PCA | 0.849 | 0.906 | 0.877 |
| HDBSCAN/UMAP | **0.936** | **0.94** | **0.938** |

Like the contrast between MDS and UMAP, the k-means clustering algorithm works directly on the Euclidean distances between data points, whereas HDBSCAN operates on a graph-based transform of the input data (Fig 9). Briefly, HDBSCAN first defines a 'mutual reachability' distance between elements, a measure of the distance between points in the dataset weighted by the local sparsity/density of each point (measured as the distance to a $k$th nearest neighbor). HDBSCAN then builds a graph, where each edge between vertices (points in the dataset) is the mutual reachability between those points, and then prunes the edges to construct a minimum spanning tree (a graph containing the minimum set of edges needed to connect all of the vertices). The minimum spanning tree is converted into a hierarchy of clusters of points sorted by mutual reachability distance, and then condensed iteratively into a smaller hierarchy of putative clusters. Finally, clusters are chosen as those that persist and are stable over the greatest range in the hierarchy.

To make the k-means algorithm more competitive with HDBSCAN, we set the number of clusters in k-means equal to the number of clusters in the hand-clustered dataset, while HDBSCAN was not parameterized at all. We computed the similarity between hand and algorithmically labeled datasets using three related metrics, homogeneity, completeness, and V-measure ([68]; see Methods section). Homogeneity measures the extent to which algorithmic clusters fall into the same hand-labeled syllable category while completeness measures the extent to which hand-labeled categories belong to the same algorithmic cluster. V-measure is the harmonic mean between the homogeneity and completeness, which is equal to the mutual information between the algorithmic clusters and the hand-labels, normalized by the mean of their marginal entropy [68].

For all three datasets, the HDBSCAN clusters most closely match those of humans as is indicated by the V-measure (Table 1). In both the Nicholson [63] Bengalese finch dataset and the Cassin's vireo dataset, the closest match to human clustering is achieved by HDBSCAN on the UMAP projections. In the Koumura dataset [64], HDBSCAN on the PCA projections gives the closest match to human clustering, where homogeneity is higher with HDBSCAN/UMAP and completeness is higher with HDBSCAN/PCA. A high homogeneity and low

completeness score indicates that algorithmic clusters tend to fall into the same hand-labeled category, but multiple sub-clusters are found within each hand labeled category. As we show in Abstracting and visualizing sequential organization, this difference between algorithmically found labels often reflects real structure in the dataset that human labeling ignores. More broadly, our clustering results show that latent projections facilitate unsupervised clustering of vocal elements into human-like syllable categories better than spectrographic representations alone. At the same time, unsupervised latent clustering is not strictly equivalent to hand labeling, and the two methods may yield different results.

**Comparing latent features and clusters to known feature spaces.** When the features underlying behaviorally relevant vocal variability in a species are known *a priori*, latent feature spaces learned directly from the data may be unnecessary to infer the underlying structure of a vocal repertoire. Although sets of behaviorally relevant features are not known for most species, Swamp sparrows are an exception, as their vocalizations have a relatively long history of careful characterization [53, 69]. Swamp sparrows produce songs that are hierarchically organized into syllables made up of shorter notes, which in turn can be well-described by only a few simple features. This set of known *a priori* features provides a useful comparison for the latent features learned by UMAP.

We compared the features learned by UMAP with the known feature-space of swamp sparrow notes using a dataset of songs recorded in the wild. In Fig 11 we show UMAP and known-feature spaces for notes from a population of swamp sparrows recorded in Conneaut Marsh, Pennsylvania. We compare the spectrogram of each note projected into UMAP space to the same note projected onto three features known to describe much of the behaviorally relevant variance in swamp sparrow song [53, 69]: peak frequency at the beginning and ending of the note (Fig 11A), note length (Fig 11B), and the overall change in peak frequency (Fig 11B). We then clustered the UMAP projections (Fig 11C) using HDBSCAN and the known feature space using a Gaussian Mixture Model (GMM; see Clustering vocalizations). For comparison, we also visualize the known features projected into UMAP (Fig 11D).

HDBSCAN found 12 unique clusters, as opposed to the normal 6-10 note categories typically used to define swamp sparrow song [53]. The GMM was set to find 10 clusters, as was used in the same dataset in prior work [53]. Between the GMM and HDBSCAN clustering, we find a degree of overlap well above chance (homogeneity = 0.633; completeness = 0.715, V-measure = 0.672; chance V-measure = 0.001; bootstrapped $p < 10^{-4}$; Fig 11A and 11B). Using the position of the note within each syllable as a common reference (most syllables were comprised of 3 or fewer notes), we compared the overlap between the two clustering methods. Both labeling schemes were similarly related to the position of notes within a syllable (e.g. first, second, third; v-measure GMM = 0.162; V-measure HDBSCAN = 0.144), and both were well above chance (bootstrapped $p < 10^{-4}$). We repeated the same analysis on a second population of swamp sparrow recorded in Hudson Valley, NY (S4 Fig), and found a similar overlap between the two clustering schemes (homogeneity = 0.643; completeness = 0.815, V-measure = 0.719; chance V-measure = 0.002; bootstrapped $p < 10^{-4}$) and a similar level of overlap with the position of notes (V-measure GMM = 0.133; V-measure HDBSCAN = 0.144).

Given this pattern of results, it is unlikely that one would want to substitute the unsupervised latent features for the known features when trying to describe swamp sparrow song in the most efficient low-dimensional space. Still, both feature sets yield surprisingly similar compressed representations. Thus, in the absence of known features, the unsupervised methods can provide either (1) a useful starting point for more refined analyses to discover "known" features, or (2) a functional analysis space that likely captures much (but not all) of the behaviorally relevant signal variation.

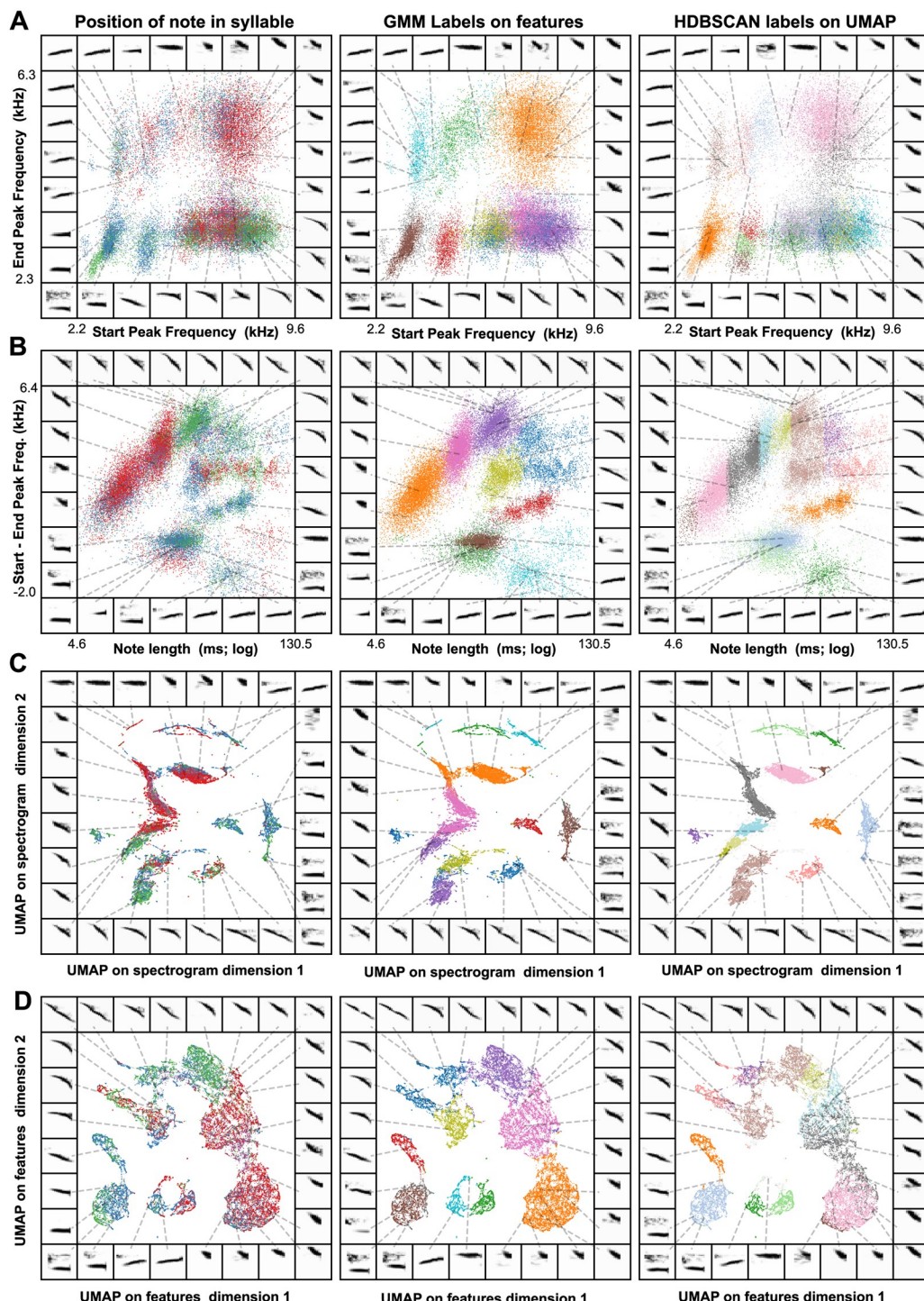

**Fig 11. Comparing latent and known features in swamp sparrow song.** (A) A scatterplot of the start and end peak frequencies of the notes produced by birds recorded in Conneaut Marsh, PA. The left panel shows notes colored by the position of each note in the syllable (red = first, blue = second, green = third). The center panel shows the sample scatterplot colored by a Gaussian Mixture Model labels (fit to the start and end peak frequencies and the note duration). The right panel shows the scatterplot colored by HDBSCAN labels over a UMAP projection of the spectrograms of notes. (B) The same notes, plotting the change in peak frequency over the note against the note's duration. (C) The same notes plotted as a UMAP projection over note-spectrograms. (D) The features from (A) and (B) projected together into a 2D UMAP space.

**Abstracting and visualizing sequential organization.**   As acoustic signals, animal vocalizations have an inherent temporal structure that can extend across time scales from short easily discretized elements such as notes, to longer duration syllables, phrases, songs, bouts, etc. The latent projection methods described above can be used to abstract corpora of song elements well-suited to temporal pattern analyses [3], and to make more direct measures of continuous vocalization time series. Moreover, their automaticity enables the high throughput necessary to satisfy intensive data requirements for most quantitative sequence models.

In practice, modeling sequential organization can be applied to any discrete dataset of vocal elements, whether labeled by hand or algorithmically. Latent projections of vocal elements have the added benefit of allowing visualization of the sequential organization that can be compared to abstracted models. As an example of this, we derived a corpus of symbolically segmented vocalizations from a dataset of Bengalese finch song using latent projections and clustering (Fig 12). Bengalese finch song bouts comprise a small number (~5-15) of highly stereotyped syllables produced in well-defined temporal sequences a few dozen syllables long [4]. We first projected syllables from a single Bengalese finch into UMAP latent space, then visualized transitions between vocal elements in latent space as line segments between points (Fig 12B), revealing highly regular patterns. To abstract this organization to a grammatical model, we clustered latent projections into discrete categories using HDBSCAN. Each bout is then treated as a sequence of symbolically labeled syllables (e.g. $B \rightarrow B \rightarrow C \rightarrow A$; Fig 12D) and the entire dataset rendered as a corpus of transcribed song (Fig 12E). Using the transcribed corpus, one can abstract statistical and grammatical models of song, such as the Markov model shown in Fig 12C or the information-theoretic analysis in Sainburg et al., [3].

**Sequential organization is tied to labeling method.**   As noted previously, hand labels and latent cluster labels of birdsong syllables generally overlap (e.g. Fig 10), but may disagree for a sizable minority of syllables (Table 1). Similarly, in mice, different algorithmic methods for abstracting and transcribing mouse vocal units (USVs) can result in substantial differences between syntactic descriptions of sequential organization [57]. We were interested in the differences between the abstracted sequential organization of birdsong when syllables were labeled by hand versus clustered in latent space. Because we have Bengalese finch datasets that are hand transcribed from two different research groups [8, 63], these datasets are ideal for comparing the sequential structure of algorithmic versus hand-transcribed song.

To contrast the two labeling methods, we first took the two Bengalese finch song datasets, projected syllables into UMAP latent space, and visualized them using the hand transcriptions provided by the datasets (Fig 13A and 13H). We then took the syllable projections and clustered them using HDBSCAN. In both datasets, we find that many individual hand-transcribed syllable categories are comprised of multiple HDBSCAN-labelled clusters in latent space (Fig 13A, 13B, 13H and 13I). To compare the different sequential abstractions of the algorithmically transcribed labels and the hand transcribed labels, we visualized the transitions between syllables in latent space (Fig 13C and 13J). These visualizations reveal that different algorithmically-transcribed clusters belonging to the same hand-transcribed label often transition to and from separate clusters in latent space. That is, the sub-category acoustics of the elements predict and are predicted by specific transitions. We visualize this effect more explicitly in Fig 13D and 13K, showing the first-order (incoming and outgoing) transitions between one hand-labeled syllable category (from Fig 13A and 13H), colored by the multiple HDBSCAN clusters that it comprises (from Fig 13B and 13I). Thus, different HDBSCAN labels that belong to the same hand-labeled category can play a different role in song-syntax, having different incoming and outgoing transitions. In Fig 13E, 13F, 13L and 13M, this complexity plays out in an abstracted Markov model, where the HDBSCAN-derived model reflects the latent transitions observed in Fig 13C and 13J more explicitly than the model abstracted from hand-labeled

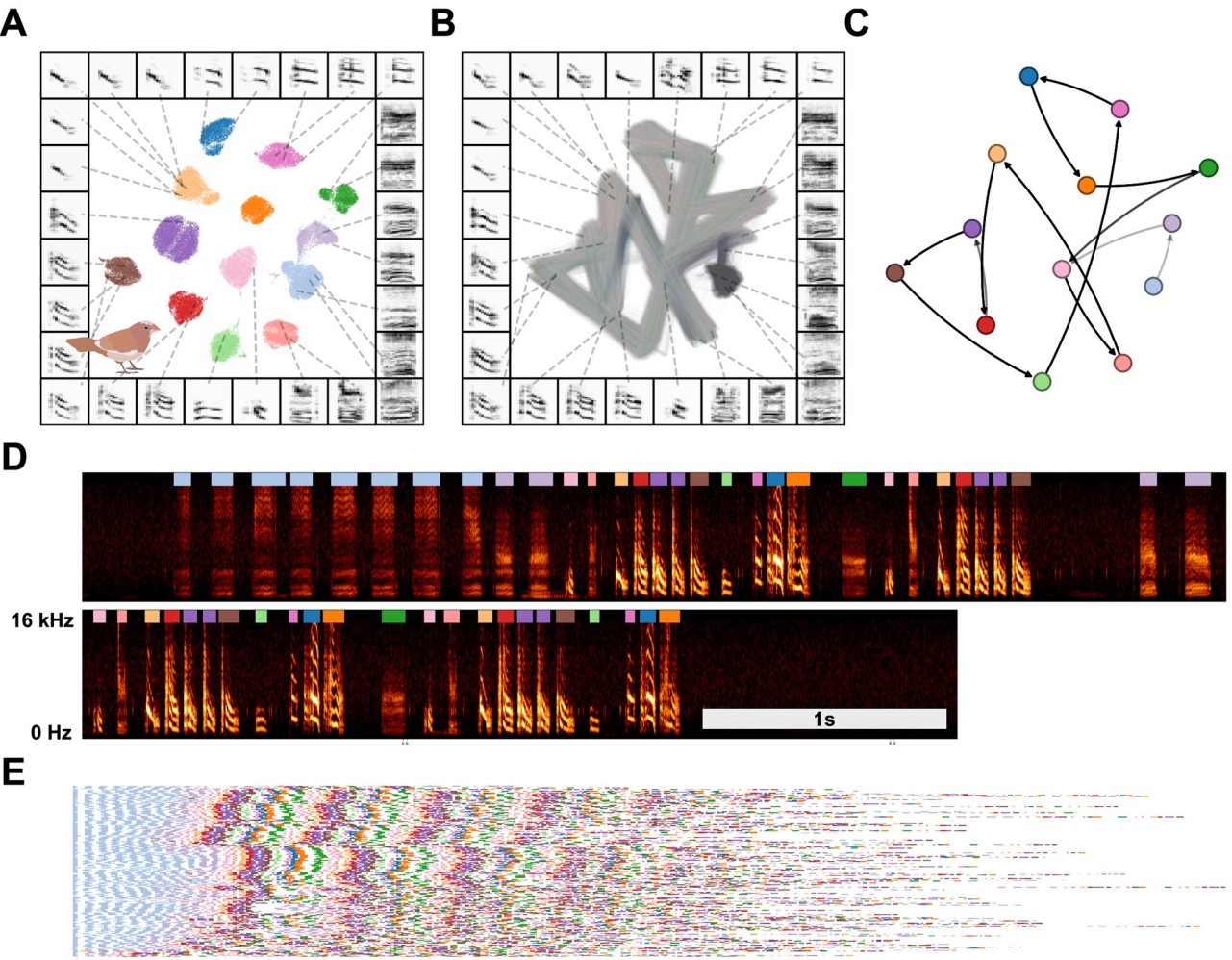

**Fig 12. Latent visualizations of Bengalese finch song sequences.** (A) Syllables of Bengalese finch songs from one individual are projected into 2D UMAP latent space and clustered using HDBSCAN. (B) Transitions between elements of song are visualized as line segments, where the color of the line segment represents its position within a bout. (C) The syllable categories and transitions in (A) and (B) can be abstracted to transition probabilities between syllable categories, as in a Markov model. (D) An example vocalization from the same individual, with syllable clusters from (A) shown above each syllable. (E) A series of song bouts. Each row is one bout, showing overlapping structure in syllable sequences. Bouts are sorted by similarity to help show structure in song.

syllables. To further understand why these clusters are labeled as the same category by hand but different categories using HDBSCAN clustering, we show example syllables from each cluster Fig 13G and 13N. Although syllables from different HDBSCAN clusters look very similar, they are differentiated by subtle yet systematic variation. Conversely, different subsets of the same experimenter-labeled category can play different syntactic roles in song sequences. The syntactic organization in Bengalese finch song is often described using Partially Observable Markov Models (POMMs) or Hidden Markov Models (HMMs), where the same syllable category plays different syntactic roles dependent on its current position in song syntax [4]. In so far as the sequential organization abstracted from hand labels obscures some of the sequential structure captured by algorithmic transcriptions, our results suggest that these different syntactic roles may be explained by the presence of different syllable categories.

To compare the difference in sequential organization captured by hand labels versus HDBSCAN labels quantitatively, we treated both HDBSCAN and hand labels as hidden states

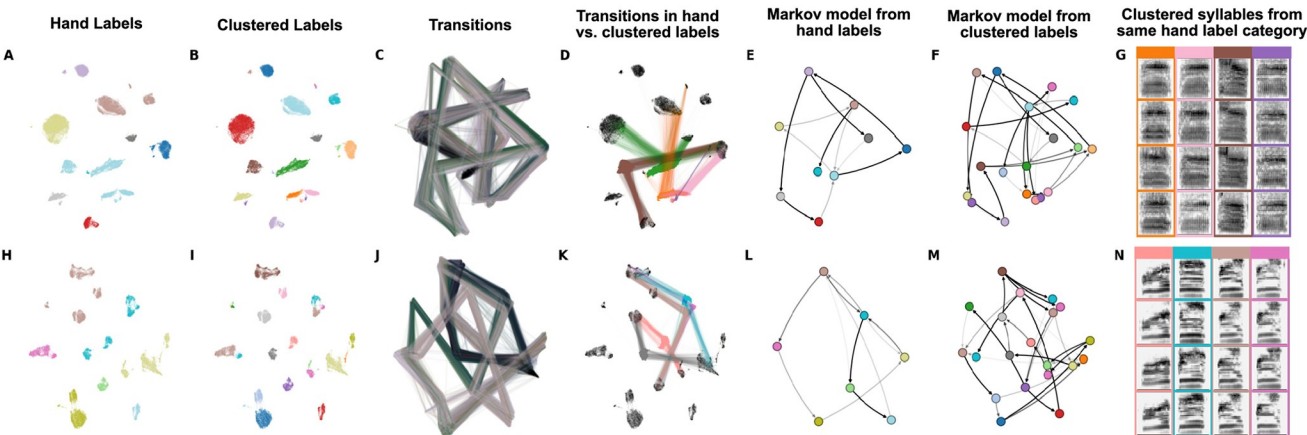

**Fig 13. Latent comparisons of hand- and algorithmically-clustered Bengalese finch song.** A-G are from a dataset produced by Nicholson et al., [9] and H-N are from a dataset produced by Koumura et al., [10] (A,H) UMAP projections of syllables of Bengalese finch song, colored by hand labels. (B,I) Algorithmic labels (UMAP/HDBSCAN). (C, J) Transitions between syllables, where color represents time within a bout of song. (D,K) Comparing the transitions between elements from a single hand-labeled category that comprises multiple algorithmically labeled clusters. Each algorithmically labeled cluster and the corresponding incoming and outgoing transitions are colored. Transitions to different regions of the UMAP projections demonstrate that the algorithmic clustering method finds clusters with different syntactic roles within hand-labeled categories. (E,L) Markov model from hand labels colored the same as in (A,H) (F,M) Markov model from clustered labels, colored the same as in (B,I). (G,N) Examples of syllables from multiple algorithmic clusters falling under a single hand-labeled cluster. Colored bounding boxes around each syllable denotes the color category from (D,K).

in separate HMMs and compared their ability to accurately model song sequences. An HMM is a finite-state model for a sequence of visible states (e.g song syllables), that is assumed to emerge from set of unobserved ('hidden') states, inferred algorithmically. To make our HMMs directly comparable, we use the hand labels as visible states, and infer hidden states from either the hand labels (e.g. Fig 14A) or the HDBSCAN labels (e.g. Fig 14B). By design, the hidden states of these two HMMs are explicitly constrained to either the hand or HDBSCAN labels, and thus ignore higher-order transitions that might carry useful sequence information. For comparison, we also trained an HMM where hidden states were inferred using the Baum-Welch algorithm and allowed to incorporate higher-order syllable sequences (e.g. Fig 14C; see Methods). For example, in the sequence of visible states $a \rightarrow b \rightarrow c \rightarrow d \rightarrow e$, there might be a hidden state representing $d|a, b, c$. HMMs allowing high-order latent representations have been used to model sequential organization in birdsong [70] and have a long history of modeling human speech.

We compared each model on its ability to predict the sequence of hand labels using the Akaike Information Criterion (AIC), which normalizes model likelihood by the number of parameters in the model [71]. Because models are compared on their ability to predict hand-labeled sequences, our comparison is biased toward sequential models based upon the hand-labels. Nonetheless, in 13 of 15 birds, the HDBSCAN clustered latent states better captured sequential dynamics ($\Delta$AIC > 2.0; Fig 14D). As expected, the Baum-Welch trained HMM is better able to explain the sequential organization in Bengalese finch song than either HMM constrained to use the hand or HDBSCAN labels in each bird ($\Delta$AIC > 2.0; Fig 14D). This indicates that second-order (or higher) transitions also contribute to the sequential structure of song in Bengalese finches. In Fig 14C, we overlay the hidden states learned by the complete HMM on the UMAP syllable projections of a single Bengalese finch from the Koumura dataset (an example bird from the Nicholson dataset is shown in S5 Fig). This reveals several clusters with clear, uniformly colored subregions, indicating HMM hidden states that are not captured

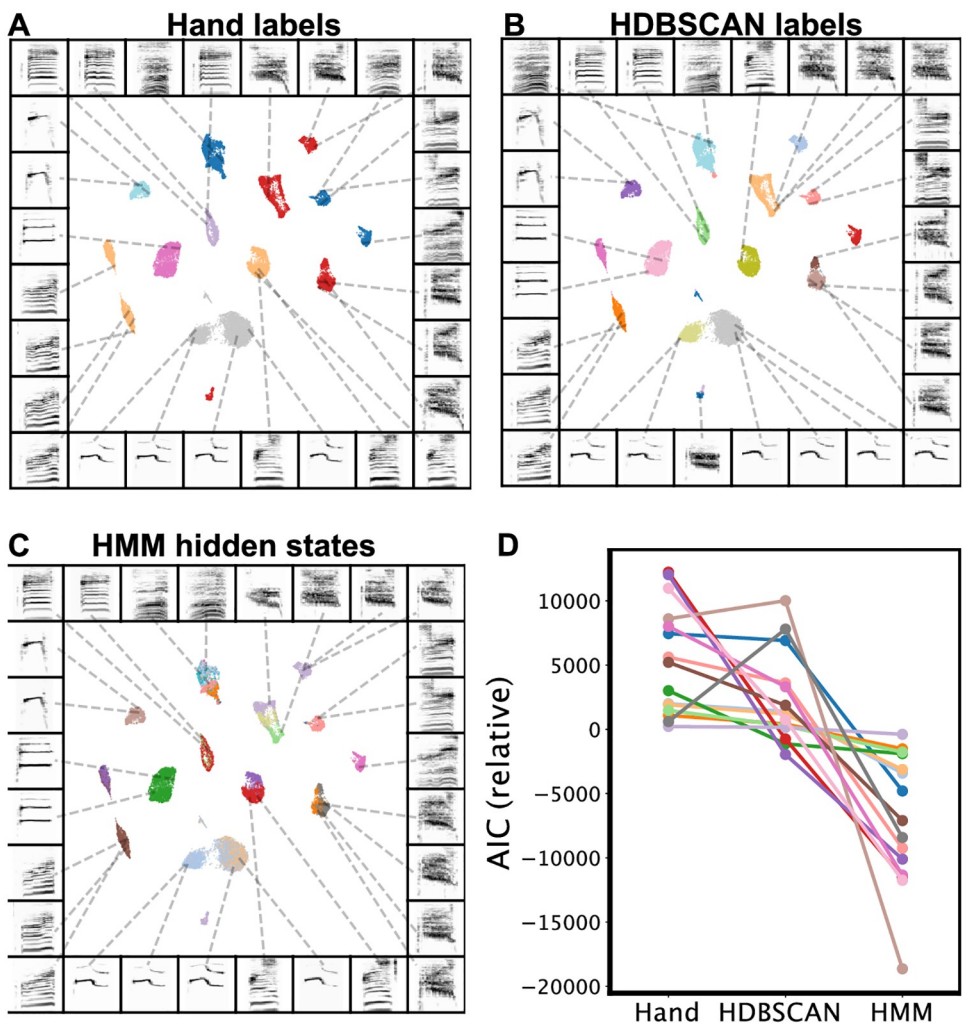

**Fig 14. Comparison of Hidden Markov Model performance using different hidden states.** Projections are shown for a single example bird from the Koumura dataset [64]. UMAP projections are labeled by three labeling schemes: (A) Hand labels (B) HDBSCAN labels on UMAP, and (C) Trained Hidden Markov Model (HMM) labels. (D) Models are compared across individual birds (points) on the basis of AIC. Each line depicts the relative (centered at zero) AIC scores for each bird for each model. Lower relative AIC equates to better model fit.

by the hand labels or HDBSCAN but still reflect non-random acoustic differences (Fig 14A and 14B).

## Temporally continuous latent trajectories

Not all vocal repertoires are made up of elements that fall into highly discrete clusters in latent space (Fig 8). For several of the datasets we analysed, categorically discrete elements are not readily apparent, making analyses such as the cluster-based analyses performed in Fig 12 more difficult. In addition, many vocalizations are difficult to segment temporally, and determining what features to use for segmentation requires careful consideration [1]. In many bird songs, for example, clear pauses exist between song elements that enable one to distinguish syllables. In other vocalizations, however, experimenters must rely on less well-defined physical features for segmentation [1, 12], which may in turn invoke a range of biases and unwarranted assumptions. At the same time, much of the research on animal vocal production, perception, and

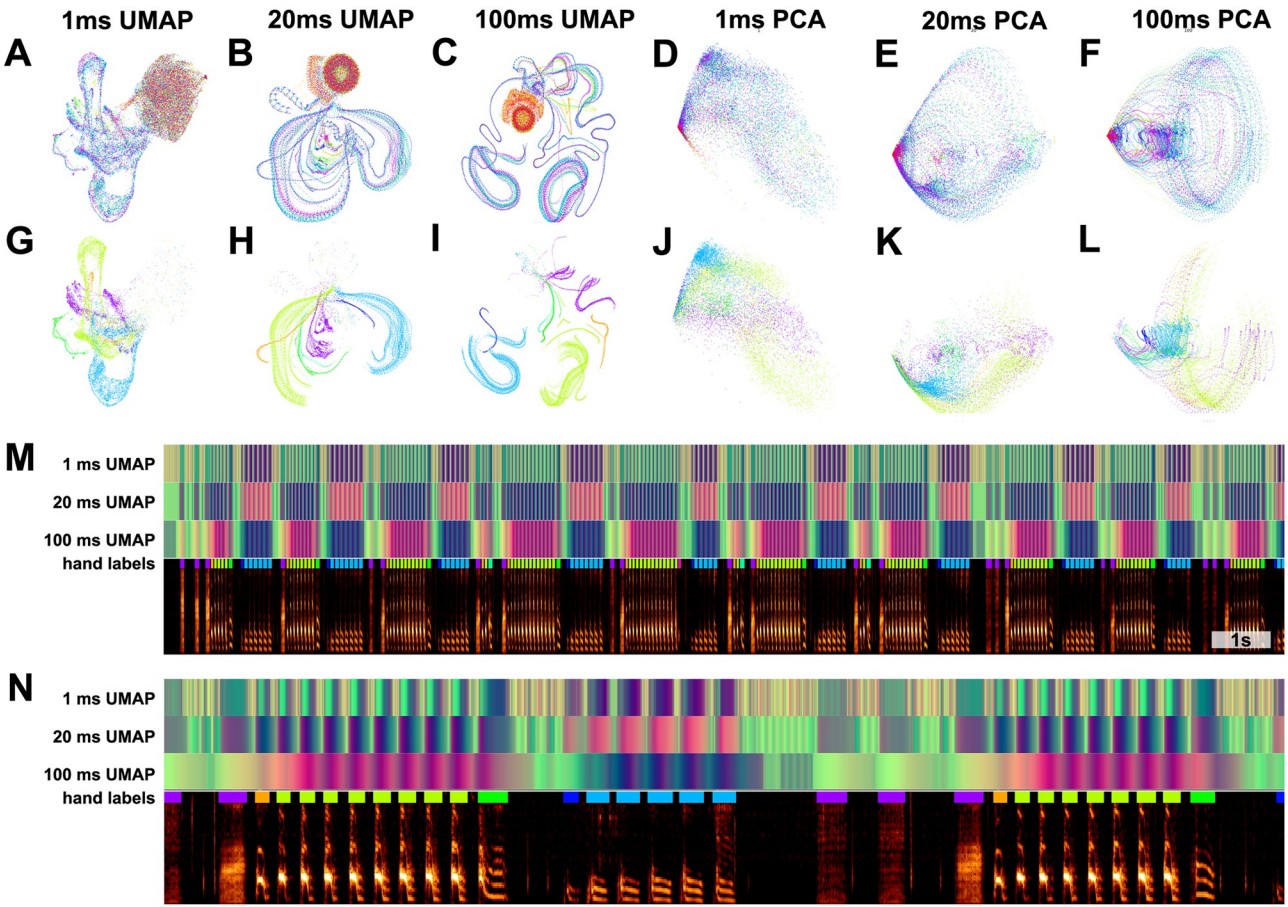

**Fig 15. Continuous UMAP projections of Bengalese finch song from a single bout produced by one individual.** (A-C) Bengalese finch song is segmented into either 1ms (A), 20ms (B), or 100ms (C) rolling windows of song, which are projected into UMAP. Color represents time within the bout of song 2(red marks the beginning and ending of the bout, corresponding to silence). (D-F) The same plots as in (A-C), projected into PCA instead of UMAP. (G-I) The same plots as (A-C) colored by hand-labeled element categories (unlabelled points are not shown). (J-L) The same plots as (D-F) colored by hand-labeled syllable categories. (M) UMAP projections represented in colorspace over a bout spectrogram. The top three rows are the UMAP projections from (A-C) projected into RGB colorspace to show the position within UMAP space over time as over the underlying spectrogram data. The fourth row are the hand labels. The final row is a bout spectrogram. (N) a subset of the bout shown in (M). In G-L, unlabeled points (points that are in between syllables) are not shown for visual clarity.

sequential organization relies on identifying "units" of a vocal repertoire [1]. To better understand the effects of temporal discretization and categorical segmentation in our analyses, we considered vocalizations as continuous trajectories in latent space and compared the resulting representations to those that treat vocal segments as single points (as in the previous Bengalese finch example in Fig 12). We explored four datasets, ranging from highly discrete clusters of vocal elements (Bengalese finch, Fig 15), to relatively discrete clustering (European starlings, Fig 16) to low clusterability (Mouse USV, Fig 17; Human speech, Fig 18). In each dataset, we find that continuous latent trajectories capture short and long timescale structure in vocal sequences without requiring vocal elements to be segmented or labeled.

**Comparing discrete and continuous representations of song in the Bengalese finch.** Bengalese finch song provides a relatively easy visual comparison between the discrete and continuous treatments of song, because it consists of a small number of unique highly stereotyped syllables (Fig 15). With a single bout of Bengalese finch song, which contains several dozen syllables, we generated a latent trajectory of song as UMAP projections of temporally-

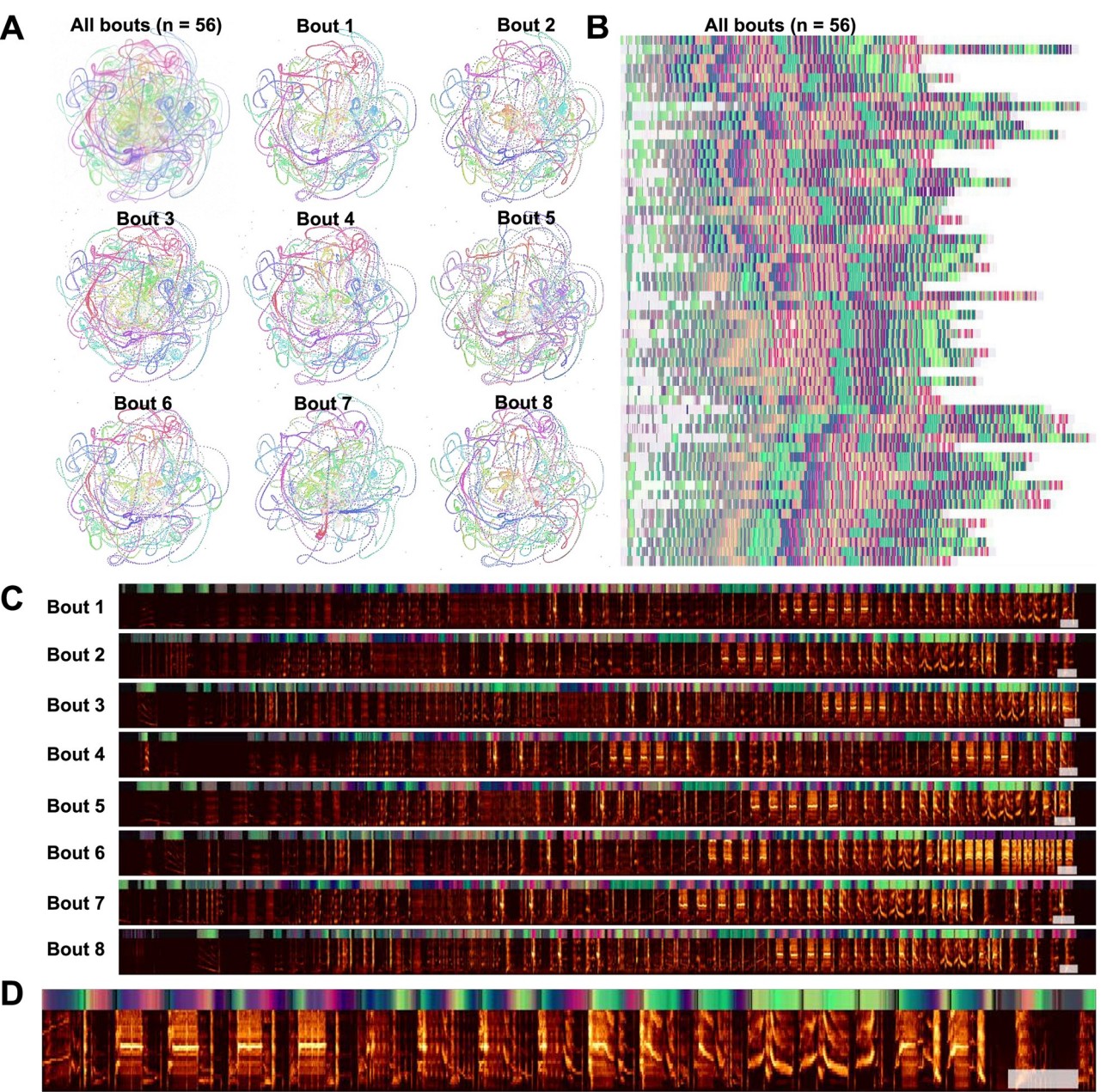

**Fig 16. Starling bouts projected into continuous UMAP space.** (A) The top left panel is each of 56 bouts of starling song projected into UMAP with a rolling window length of 200ms, color represents time within the bout. Each of the other 8 panels is a single bout, demonstrating the high similarity across bouts. (B) Latent UMAP projections of the 56 bouts of song projected into colorspace in the same manner as Fig 15M. Although the exact structure of a bout of song is variable from rendition to rendition, similar elements tend to occur at similar regions of song and the overall structure is preserved. (C) The eight example bouts from (A) with UMAP colorspace projections above. The white box at the end of each plot corresponds to one second. (D) A zoomed-in section of the first spectrogram in C.

rolling windows of the bout spectrogram (See Projections section). To explore this latent space, we varied the window length between 1 and 100ms (Fig 15A–15L). At each window size, we compared UMAP projections (Fig 15A–15C) to PCA projections (Fig 15D–15F). In both PCA and UMAP, trajectories are more clearly visible as window size increases across the range tested, and overall the UMAP trajectories show more well-defined structure than the

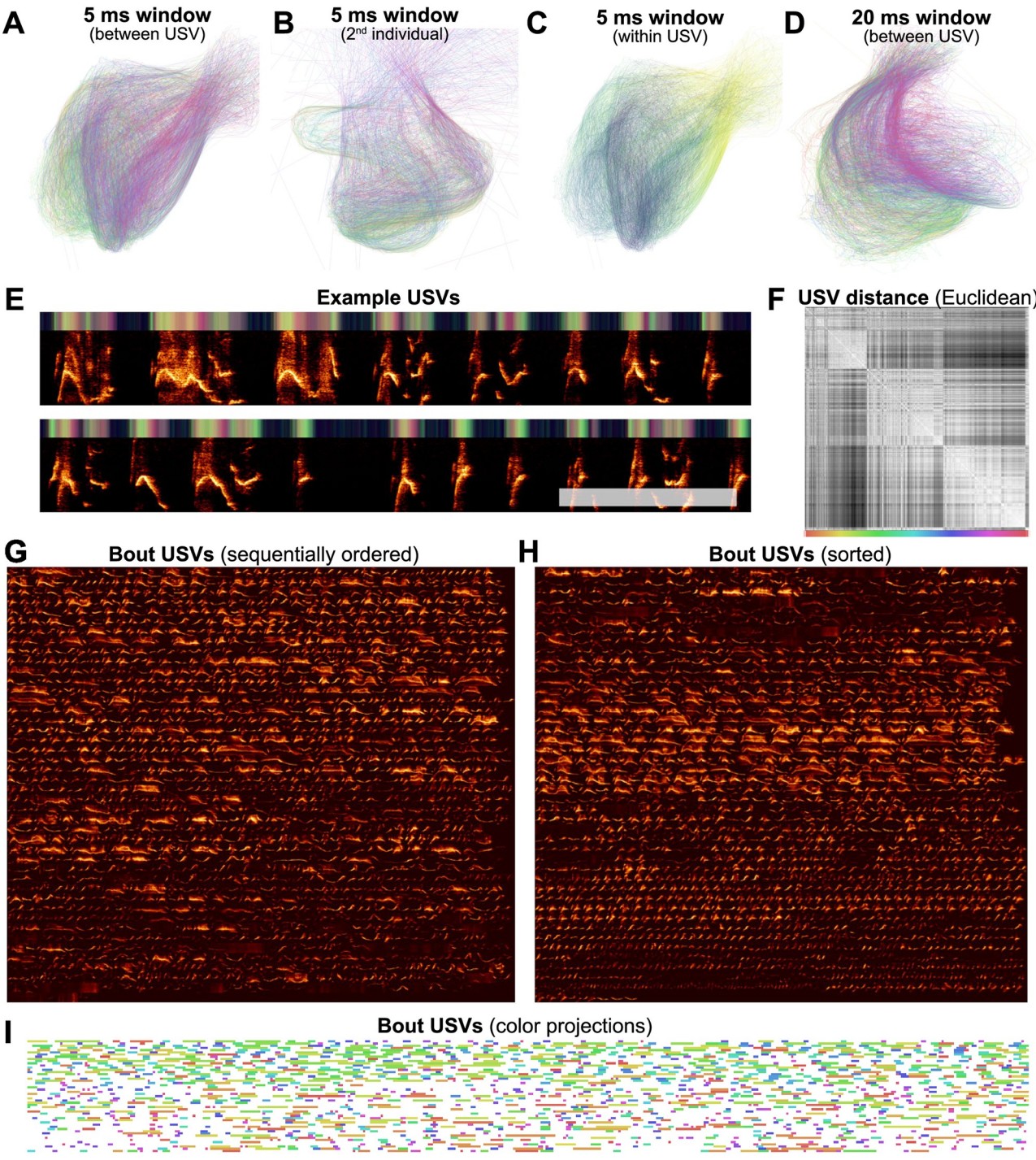

**Fig 17. USV patterns revealed through latent projections of a single mouse vocal sequence.** (A) Each USV is plotted as a line and colored by its position within the sequence. Projections are sampled from a 5ms rolling window. (B) Projections from a different recording from a second individual using the same method as in (A). (C) The same plot as in A, where color represents time within a USV. (D) The same plot as in (A) but with a 20ms rolling window. (E) An example section of the USVs from (A), where the bar on the top of the plot shows the UMAP projections in colorspace (the first and second UMAP dimensions are plotted as color dimensions). 2The white scale bar corresponds to 250ms. (F) A distance matrix between each of 1,590 USVs produced in the sequence visualized in (A), reordered so that similar USVs are closer to one another. (G) Each of the 1,590 USVs produced in the sequence from (A), in order (left to right, top to bottom). (H) The same USVs as in (G), reordered based upon the distance matrix in (F). (I) The entire sequence from (A) where USVs are color-coded based upon their position in the distance matrix in (F).

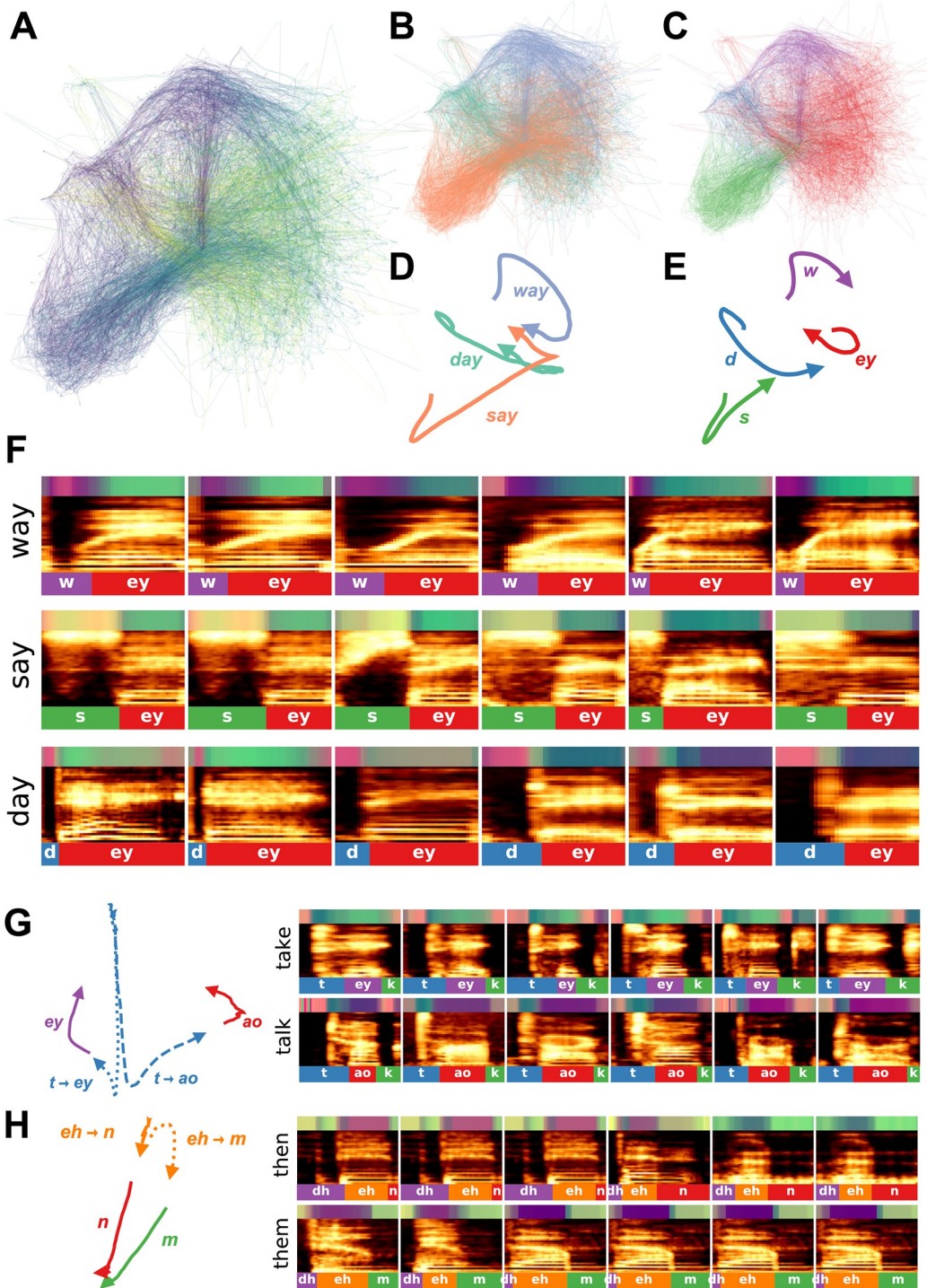

**Fig 18. Speech trajectories showing coarticulation in minimal pairs.** (A) Utterances of the words 'day', 'say', and 'way' are projected into a continuous UMAP latent space with a window size of 4ms. Color represents time, where darker is earlier in the word. (B) The same projections as in (A) but color-coded by the corresponding word. (C) The same projections are colored by the corresponding phonemes. (D) The average latent trajectory for each word. (E) The average trajectory for each phoneme. (F) Example spectrograms of words, with latent trajectories above spectrograms and phoneme labels below spectrograms. (G) Average trajectories and corresponding spectrograms for the words 'take' and 'talk' showing the different trajectories for 't' in each word. (H) Average trajectories and the corresponding spectrograms for the words 'then' and 'them' showing the different trajectories for 'eh' in each word.

PCA trajectories. To compare continuous projections to discrete syllables, we re-colored the continuous trajectories by the discrete syllable labels obtained from the dataset. Again, as the window size increases, each syllable converges to a more distinct trajectory in UMAP space (Fig 15G–15I). To visualize the discrete syllable labels and the continuous latent projections in relation to song, we converted the 2D projections into colorspace and show them as a continuous trajectory alongside the song spectrograms and discrete labels in Fig 15M and 15N. Colorspace representations of the 2D projections consist of treating the two UMAP dimensions as either a red, green, or blue channel in RGB (3D) colorspace, and holding the third channel constant. This creates a colormap projection of the two UMAP dimensions.

**Latent trajectories of European starling song.** European starling song provides an interesting case study for exploring the sequential organization of song using continuous latent projections because starling song is more sequentially complex than Bengalese finch song, but is still highly stereotyped and has well-characterized temporal structure. European starling song is comprised of a large number of individual song elements, usually transcribed as 'motifs', that are produced within a bout of singing. Song bouts last several tens of seconds and contain many unique motifs grouped into three broad classes: introductory whistles, variable motifs, and high-frequency terminal motifs [72]. Motifs are variable within classes, and variability is affected by the presence of potential mates and seasonality [73, 74]. Although sequentially ordered motifs are usually segmentable by gaps of silence occurring when starlings are taking breaths, segmenting motifs using silence alone can be difficult because pauses are often short and bleed into surrounding syllables [75]. When syllables are temporally discretized, they are relatively clusterable (Fig 8), however syllables tend to vary somewhat continuously (Fig 16D). To analyze starling song independent of assumptions about segment (motif) boundaries and element categories, we projected bouts of song from a single male European starling into UMAP trajectories using the same methods as with Bengalese finch song in Fig 15. We used a 200ms time window for these projections, around the order of a shorter syllable of starling song and longer than the pause in between syllables, resulting our projections capturing information about transitions between syllables. Time windows of different lengths reveal structure at different timescales, for example windows shorter than the length of a pause between syllables will return to the region of latent space corresponding to silence (e.g. Fig 15A and 15B) and capture within syllable structure but not the transitions between syllables.

We find that the broad structure of song bouts are highly repetitive across renditions, but contain elements within each bout that are variable across bout renditions. For example, in Fig 16A, the top left plot is an overlay showing the trajectories of 56 bouts performed by a single bird, with color representing time within each bout. The eight plots surrounding it are single bout renditions. Different song elements are well time-locked as indicated by a similar hue present in the same regions of each plot. Additionally, most parts of the song occur in each rendition. However, certain song elements are produced or repeated in some renditions but not others. To illustrate this better, in Fig 16B, we show the same 56 bouts projected into colorspace in the same manner as Fig 15M and 15N, where each row is one bout rendition. We observe that, while each rendition contains most of the same patterns at relatively similar times, some patterns occur more variably. In Fig 16C and 16D we show example spectrograms corresponding to latent projections in Fig 16A, showing how the latent projections map onto spectrograms.

Quantifying and visualizing the sequential structure of song using continuous trajectories rather than discrete element labels is robust to errors and biases in segmenting and categorizing syllables of song. Our results show the potential utility of continuous latent trajectories as a viable alternative to discrete methods for analyzing song structure even with highly complex, many-element, song.

**Latent trajectories and clusterability of mouse USVs.**   House mice produce ultrasonic vocalizations (USVs) comprising temporally discrete syllable-like elements that are hierarchically organized and produced over long timescales, generally lasting seconds to minutes [76]. When analyzed for temporal structure, mouse vocalizations are typically segmented into temporally-discrete USVs and then categorized into discrete clusters [1, 57, 76–78] in a manner similar to syllables of birdsong. As was observed on the basis of the Hopkin's statistic (Fig 8), however, USVs do not cluster into discrete distributions in the same manner as birdsong. Choosing different arbitrary clustering heuristics will therefore have profound impacts on downstream analyses of sequential organization [57].

We sought to better understand the continuous variation present in mouse USVs, and explore the sequential organization of mouse vocalizations without having to categorize USVs. To do this, we represented mouse USVs as continuous trajectories (Fig 17E) in UMAP latent space using similar methods as with starlings (Fig 16) and finches (Fig 15). In Fig 17, we use a single recording of one individual producing 1,590 (Fig 17G) USVs over 205 seconds as a case study to examine the categorical and sequential organization of USVs. We projected every USV produced in that sequence as a trajectory in UMAP latent space (Fig 17A, 17C and 17D). Similar to our observations in Fig 8I using discrete segments, we do not observe clear element categories within continuous trajectories, as observed for Bengalese finch song (e.g. Fig 15I).

To explore the categorical structure of USVs further, we reordered all of the USVs in Fig 17G by the similarity of their latent trajectories (measured by the Euclidean distance between latent projection vectors; Fig 17F) and plotted them side-by-side (Fig 17H). Both the similarity matrix of the latent trajectories (Fig 17F) and the similarity-reordered spectrograms (Fig 17H) show that while some USVs are similar to their neighbors, no highly stereotyped USV categories are observable.

Although USVs do not aggregate into clearly discernible, discrete clusters, the temporal organization of USVs within the vocal sequence is not random. Some latent trajectories are more frequent at different parts of the vocalization. In Fig 17A, we color-coded USV trajectories according to each USV's position within the sequence. The local similarities in coloring (e.g., the purple and green hues) indicate that specific USV trajectories tend to occur in distinct parts of the sequence. Arranging all of the USVs in order (Fig 17G) makes this organization more evident, where one can see that shorter and lower amplitude USVs tend to occur more frequently at the end of the sequence. To visualize the vocalizations as a sequence of discrete elements, we plotted the entire sequence of USVs (Fig 17I), with colored labels representing the USV's position in the reordered similarity matrix (in a similar manner as the discrete category labels in Fig 15E). In this visualization, one can see that different colors dominate different parts of the sequence, again reflecting that shorter and quieter USVs tend to occur at the end of the sequence.

**Latent trajectories of human speech.**   Discrete elements of human speech (i.e. phonemes) are not spoken in isolation and their acoustics are influenced by neighboring sounds, a process termed co-articulation. For example, when producing the words 'day', 'say', or 'way', the position of the tongue, lips, and teeth differ dramatically at the beginning of the phoneme 'ey' due to the preceding 'd', 's', or 'w' phonemes, respectively. This results in differences in the pronunciation of 'ey' across words (Fig 18F). Co-articulation explains much of the acoustic variation observed within phonetic categories. Abstracting to phonetic categories therefore discounts much of this context-dependent acoustic variance.

We explored co-articulation in speech, by projecting sets of words differing by a single phoneme (i.e. minimal pairs) into continuous latent spaces, then extracted trajectories of words and phonemes that capture sub-phonetic context-dependency (Fig 18). We obtained the words from the same Buckeye corpus of conversational English used in Figs 8 and 7, and

S3 Fig. We computed spectrograms over all examples of each target word, then projected sliding 4-ms windows from each spectrogram into UMAP latent space to yield a continuous vocal trajectory over each word (Fig 18). We visualized trajectories by their corresponding word and phoneme labels (Fig 18B and 18C) and computed the average latent trajectory for each word and phoneme (Fig 18D and 18E). The average trajectories reveal context-dependent variation within phonemes caused by coarticulation. For example, the words 'way', 'day', and 'say' each end in the same phoneme ('ey'; Fig 18A–18F), which appears as an overlapping region in the latent space (the red region in Fig 18C). The endings of each average word trajectory vary, however, indicating that the production of 'ey' differs based on its specific context (Fig 18D). The difference between the production of 'ey' can be observed in the average latent trajectory over each word, where the trajectories for 'day' and 'say' end in a sharp transition, while the trajectory for 'way' is smoother (Fig 18D). These differences are apparent in Fig 18F which shows examples of each word's spectrogram accompanied by its corresponding phoneme labels and color-coded latent trajectory. In the production of 'say' and 'day' a more abrupt transition occurs in latent space between 's'/'d' and 'ey', as indicated by the yellow to blue-green transitions above spectrograms in 'say' and the pink to blue-green transition above 'day'. For 'way', in contrast, a smoother transition occurs from the purple region of latent space corresponding to 'w' to the blue-green region of latent space corresponding to 'ey'.

Latent space trajectories can reveal other co-articulations as well. In Fig 18G, we show the different trajectories characterizing the phoneme 't' in the context of the word 'take' versus 'talk'. In this case, the 't' phoneme follows a similar trajectory for both words until it nears the next phoneme ('ey' vs. 'ao'), at which point the production of 't' diverges for the different words. A similar example can be seen for co-articulation of the phoneme 'eh' in the words 'them' versus 'then' (Fig 18H). These examples show the utility of latent trajectories in describing sub-phonemic variation in speech signals in a continuous manner rather than as discrete units.

## Discussion

We have presented a set of computational methods for projecting vocal communication signals into low-dimensional latent representational spaces, learned directly from the spectrograms of the signals. We demonstrate the flexibility and power of these methods by applying them to a wide sample of animal vocal communication signals, including songbirds, primates, rodents, bats, and cetaceans (Fig 8). Deployed over short timescales of a few hundred milliseconds, our methods capture significant behaviorally-relevant structure in the spectro-temporal acoustics of these diverse species' vocalizations. We find that complex attributes of vocal signals, such as individual identity (Fig 4), species identity (Fig 5A and 5B), geographic population variability (Fig 5C), phonetics (Fig 7, S3 Fig), and similarity-based clusters (Fig 10) can all be captured by the unsupervised latent space representations we present. We also show that songbirds tend to produce signals that cluster discretely in latent space, whereas mammalian vocalizations are more uniformly distributed, an observation that deserves much closer investigation in more species. Applied to longer timescales, spanning seconds or minutes, the same methods allowed us to visualize sequential organization and test models of vocal sequencing (Fig 12). We demonstrated that in some cases latent approaches confer advantages over hand labeling or supervised learning (Figs 13 and 14). Finally, we visualized vocalizations as continuous trajectories in latent space (Figs 15, 16, 17 and 18), providing a powerful method for studying sequential organization without discretization [1].

Latent models have shown increasing utility in the biological sciences over the past several years. As machine learning algorithms improve, so will their utility in characterizing the

complex patterns present in biological systems like animal communication. In neuroscience, latent models already play an important role in characterizing complex neural population dynamics [40]. Similarly, latent models are playing an increasingly important role in computational ethology [17], where characterizations of animal movements and behaviors have uncovered complex sequential organization [38, 79, 80]. In animal communication, pattern recognition using various machine learning techniques has been used to characterize vocalizations and label auditory objects [3, 34, 35, 39, 57, 77, 78]. Our work furthers this emerging research area by demonstrating the utility of unsupervised latent models for both systematically visualizing and abstracting structure from animal vocalizations across a wide range of species.

## Latent and known features

Our methods show that *a priori* feature-based compression is not a prerequisite to progress in understanding behaviorally relevant acoustic diversity. The methods we describe are not meant, however, as a wholesale replacement of more traditional analyses based on compression of vocal signals into known behaviorally-relevant feature spaces. In most cases, these known feature spaces are the result of careful exploration, experimentation, and testing, and therefore encapsulate an invaluable pool of knowledge. When available, this knowledge should be used. Our methods are most useful when this knowledge is either unavailable, or may not hold for all of the species one wishes to investigate. For comparison across species, they provide a common space that is unbiased by the features of any one species (Fig 5), and within species they can reveal behaviorally relevant structure (Figs 4, 6 and 7). In the cases where we compared latent features to the representations of signals based on known features (Figs 3 and 11), it is clear that the known features captured aspects of the signals that the latent representations missed. At the same time, however, the latent representations capture much of the same variance, albeit without reference to intuitive features. Thus, when possible the distributional properties of signals revealed by our unsupervised methods can (and should be) linked to specific physical features of the signals.

In light of the observation that latent features can provide a close approximation to feature-based representations, it is interesting to ask why UMAP works as well as it does in the myriad ways we have shown. Like other compression algorithms, UMAP relies on statistical regularities in the input data to find the low dimensional manifold that best captures a combination of global and local structure. Thus, the co-variance between behaviorally relevant features and those revealed in UMAP indicates that behaviorally relevant dimensions of signals contain reliable acoustic variance. In other words, the statistical structure of the signals reflect their function. While these may not be the dimensions of maximal variance over the whole signal set (which is why PCA can miss them), the local variance is reliable enough to be captured by UMAP. While it may be less surprising to note that animal communication relies upon reliable signal variance to convey information, it is noteworthy that our signal analysis methods have advanced to the point where we can directly measure that variance without prior knowledge.

## Discrete and continuous representations of vocalizations

Studies of animal communication classically rely on segmenting vocalizations into discrete temporal units. In many species, this temporal segmentation is a natural step in representing and analyzing vocal data. In birdsong, for example, temporally distinct syllables are often well defined by clear pauses between highly stereotyped syllable categories (Fig 8O). We showed that the syllables labeled through unsupervised clustering account for sequential organization in Bengalese finch song better than experimenter-defined hand labels, even when hand-labels

are treated as ground-truth (Fig 14D). Using an HMM that was free to define states based on higher-order sequential dynamics revealed even finer sub-classes of elements with reliable acoustic structure (Fig 14D). Thus, one strategy to improve syllable labeling algorithms going forward is to include models for the sequential dynamics of vocalizations (e.g [81]). Such models should take into account recent findings that Markovian assumptions do not fully account for the long-range dynamics in all bird songs [3] or other signals with long-range organization such as human speech [82]. Lastly, neither density-based clustering, hand clustering, nor sequence-based clustering, model the animal's categorical perception directly. Therefore, making perceptual inferences based upon these labels is limited without behavioral or physiological investigations with the animal.

Another strategy for studying vocal sequences is to avoid the problem of segmentation/discretization altogether. Indeed, in many non-avian species, vocal elements are either not clearly stereotyped or temporally distinct (Fig 8), and methods for segmentation can vary based upon changes in a range of acoustic properties, similar sounds, or higher-order organization [1]. These constraints force experimenters to make decisions that can affect downstream analyses [35, 57]. We projected continuous latent representations of vocalizations ranging from the highly stereotyped song of Bengalese finches, to highly variable mouse USVs, and found that continuous latent projections effectively described useful aspects of spectro-temporal structure and sequential organization (Figs 15, 16 and 17). Continuous latent variable projections of human speech capture sub-phoneme temporal dynamics that correspond to co-articulation (Fig 18). Collectively, our results show that continuous latent representations of vocalizations provide an alternative to discrete segment-based representations while remaining agnostic to segment boundaries, and without the need to segment vocalizations into discrete elements or symbolic categories. Of course, where elements can be clustered into clear and discrete element categories, it may be valuable to do so. The link from temporally continuous vocalization to symbolically discrete sequences will be an important target for future investigations.

## Limitations

Throughout this manuscript, we have discussed and applied a set of tools for analyzing vocal communication signals. Although we have spent much of the manuscript focusing on the utility of these tools, there are limits to their application. Here, we discuss a few of the drawbacks and challenges.

**Unsupervised learning with noisy vocal signals.**   Supervised learning algorithms are trained explicitly to learn what features of a dataset are relevant to mapping input data to a set of labels. For example, a neural network trained to classify birdsong syllables based upon hand labels is a supervised algorithm. Such algorithms learn what parts of the data are relevant to this mapping (signal), and what parts of data should be ignored (noise). Conversely, unsupervised learning algorithms model structure in data without external reference to what is signal and what is noise. The datasets we used ranged from relatively noisy signals recorded in the wild, to recordings in a laboratory setting from a single individual in a sound-isolated chamber. Algorithms like PCA or UMAP can ignore some level of noise in data. In PCA, for example, when signal explains more variance in the data than noise, the first few principal components will capture primarily signal. Similarly, UMAP embeddings rely on the construction of a nearest neighbor graph. So long as noise does not substantially influence the construction of this graph, some degree of noise can be ignored. Still, high background noise in recordings can impact the quality of latent projections. Thus, signal-aware methods for reducing noise before projecting the data would be ideal. In the methods, we discuss one method we used to decrease background time-domain noise in some of the noisier signals using a

technique called "spectral gating". Considerations of how to reduce the noise in data are crucial to modeling structure in animal communication signals, especially using unsupervised learning algorithms.

**Unsupervised learning with small vocalization datasets.**   Contemporary machine learning algorithms often rely on very large datasets. To learn the structure of a complex vocal communicative repertoire, having more coverage over the vocal repertoire is better; it would be difficult to find clusters of vocalizations when only a few exemplars are available for each vocalization. In contrast, when features of a dataset are already known, less data is needed to make a comparison. For example, it might take a machine learning algorithm many exemplars to untangle data in such a way that important features like fundamental frequency are learned. As such, when datasets are small, methods like UMAP are less useful in modeling data, and carefully selecting features is generally a more appropriate method for making comparisons.

**Representing data and distance across vocalizations.**   Graph-based models like UMAP find structure in data by building graphical representations of datasets and then embedding those graphs into low-dimensional spaces. Building a graphical representation of a dataset is predicated on determining a notion of distance between points. Deciding how to measure the distance between two elements of animal communication requires careful thought. Throughout this manuscript, we computed spectrograms of vocalizations, and computed distance as the Euclidean distance between those spectrograms. This measure of distance, while easy to compute, is one of many ways to measure the distance between points.

The use of both PAFs and spectrograms should be considered carefully when making comparisons in vocal datasets. Descriptive statistics can be overly reductive and may not capture all of the relevant characteristics of the signal, while spectrogram representations can be overcomplete, and require further dimensionality reduction to reveal relevant features in statistical analyses [28, 83, 84]. Treating time-frequency bins of spectrograms as independent features inaccurately reflects the perceptual space of animals, who are sensitive to relative relationships between time varying components and spectral shape (e.g., [85]) less than absolute power at specific at a specific time or frequency. For example, the spectrograms of two identically shaped vocalizations shifted in frequency by a quarter octave may appear completely uncorrelated when each time-frequency coefficient is treated as an independent dimension. Yet, those same vocalizations might be treated as effectively the same by a receiver. Topological methods such as UMAP or t-SNE partially resolve this issue, because their graph-based representations rely on the relationships between neighboring data points as inputs. As a result, vocalizations that are distant in Euclidean space (i.e. whose spectrograms are uncorrelated) can be close in latent space. Even when using spectrograms, determining the parameters of the spectrogram is an important consideration, and can impact the result of downstream machine learning tasks for bioacoustics [28, 86].

Constructing a graph in UMAP relies on computing the distances between some representation of the data (here, vocalizations). Representing vocal elements as spectrograms or PAFs, and constructing a graph on the basis of the Euclidean distance between those features are two ways of constructing that graph. In principle, any distance metric could be used in place of Euclidean distance to build the graph in UMAP. For example, the distance between two spectrograms can be computed using Dynamic Time Warping (DTW) [87, 88], Dynamic Frequency Warping (DFW) [89], or peaks in cross-correlations [90] to add invariance to shifts in time and frequency between vocal elements(in the code [91], we show an example of how DTW can be used as the distance metric in UMAP instead of Euclidean distance). Determining what notion of distance is most reasonable to compare two vocalizations requires consideration. When acoustic features are known to capture the structure of an animal's communication, either by careful study or explicitly probing an animal's perceptual representations of

their vocal repertoire, the distance can be computed on the basis of those acoustic features. Here, we use Euclidean distance between spectrograms to build UMAP graphs, which we find is effective to capture structure in many vocal signals.

**Parameterization and understanding structure in latent projections.**   It has been well documented that the structure found using graph-based dimensionality reduction algorithms like t-SNE and UMAP can be heavily biased by the parameterization used in the algorithm [33, 92]. Generally, the default parameters used in UMAP are good starting points. In this manuscript, we used the default parameters in all of our projections, except where otherwise noted. Still, exploring the persistence of structure across parameterizations is an important consideration when making inferences based upon structure in latent space.

## Future work

**Synthesizing animal vocalization signals.**   The present work discusses latent models from the angle of dimensionality reduction, learning a low dimensional descriptive representation of the structure of the signal. Here, we left a second important aspect of latent models unexplored: generativity. One aspect of machine learning that is largely under-utilized in animal communication research, and psychophysics more generally, is using generative latent models that jointly model the probability in data space and latent space, to generate vocalizations directly from samples in latent space. Generative techniques enable the synthesis of complex, high-dimensional data such as animal communication signals by sampling from low-dimensional latent spaces. Preliminary work has already been done in this area, for example, generating syllables of birdsong as stimuli for psychophysical and neurophysiological probes [36, 37] using deep neural networks. HMMs have also been used to synthesize vocalizations [93]. With the recent advancements in machine learning, especially in areas such as generative modeling and text-to-speech, the synthesis of high-fidelity animal vocal signals is likely to become an important avenue for studying the full spectrum of vocal communication in more biologically realistic ways.

**Local and global structure.**   The methods we present in this paper center around the graph-based dimensionality reduction algorithm UMAP. Graph-based dimensionality reduction algorithms like t-SNE and UMAP favor the preservation of local structure of global structure, as opposed to PCA and MDS, which favor the preservation of global structure. Capturing local structure means mapping nearby points in data-space to nearby points in the low-dimensional embedding space, while capturing global structure means preserving relationships at all scales; both local and more distant [94]. UMAP and t-SNE capture much more local structure than PCA, but less global structure. However, UMAP is an improvement over t-SNE in that it captures more global structure [31]. The current deficit in capturing global structure with graph-based dimensionality reduction algorithms is not necessarily a fundamental issue, however. Future advancements in non-linear graph-based dimensionality reduction algorithms will likely better capturing global structure. Capturing the density of distributions (like clusters of birdsong elements) is also likely an important feature of dimensionality reduction algorithms. At present, neither UMAP nor t-SNE embeddings are designed to capture local density (the distances between points) in data space (they are explicitly designed not to). Recent improvements on this front [95], for example, might aid in finding structural differences between directed birdsong which is more highly stereotyped and undirected birdsong, which is more exploratory. Advances in non-linear graph-based dimensionality reduction algorithms are likely to have important impacts on quantifying latent structure in vocal data.

**Further directions.**   The work presented here is a first step in exploring the potential power of latent modeling in animal communication. We touch only briefly on a number of

questions that we find interesting and think important within the field of animal communication. Other researchers may certainly want to target other questions, and we hope that some of these techniques (and the provided code) may be adapted in that service. Our analyses were taken from a diverse range of animals, sampled in diverse conditions both in the wild and in the laboratory, and are thus not well controlled for variability between species. Certainly, as bioacoustic data becomes more open and readily available, testing large, cross-species, hypotheses will become more plausible. We introduced several areas in which latent models can act as a powerful tool to visually and quantitatively explore complex variation in vocal data. These methods are not restricted to bioacoustic data, however. We hope that the work presented here will encourage a larger incorporation of latent and unsupervised modeling as a means to represent, understand, and experiment with animal communication signals in general. At present, our work exhibits the utility of latent modeling on a small sampling of the many directions that can be taken in the characterization of animal communication.

## Methods

### Datasets

The Buckeye [96] dataset of conversational English was used for human speech. The swamp sparrow dataset is from [21] and was acquired from [97]. The California thrasher dataset is from [6] and was acquired from BirdDB [41]. The Cassin's vireo dataset is from [7] and was also acquired from BirdDB. The giant otter dataset was acquired from [98]. The canary song dataset is from [5] and was acquired via personal correspondence. Two zebra finch datasets were used. The first is a dataset comprised of a large number of motifs produced by several individuals from [99]. The second is a smaller library of vocalizations with more diverse vocalization types and a greater number of individuals than the motif dataset. It correspond to data from [28] and [24] and was acquired via personal correspondence. The white-rumped munia dataset is from [4]. The humpback whale dataset was acquired from Mobysound [100]. The house mice USV dataset was acquired from [76]. An additional higher SNR dataset of mouse USVs was sent from the same group via personal correspondence. The European starling dataset is from [3] and was acquired from [101]. The gibbon song is from [102]. The marmoset dataset was received via personal correspondence and was recorded similarly to [47]. The fruit bat data is from [103] and was acquired from [104]. The macaque data is from [27] and was acquired from [105]. The beaked whale dataset is from [51] and was acquired from [106]. The North American birds dataset is from [107] and was acquired from [50]. We used two Bengalese finch datasets. The first is from [8] and was acquired from [64]. The second is from [63].

### Reducing noise in audio

One issue with automated analyses over animal communication is the requirement for signals to have relatively low background noise in their recordings. In part, background noise reduction can be performed algorithmically. At the same time, noisier data requires a greater degree of human intervention to tell the algorithm what to consider signal, and what to consider noise. While some of the datasets used in our analyses were recorded in sound-isolated chambers in a laboratory, others were recorded in nature. The datasets we ultimately used for this paper were either relatively low noise or had some hand-annotations that were necessary to determine where syllables started and ended. For example, many of the datasets had hand segmented vocal element boundaries that were used instead of algorithmic segmentation. We show a comparison of the silhouette score of the Cassin's dataset used in Fig 2 for different signal-to-noise ratios (SNR) in S6 Fig.

To reduce the background noise in acoustic signals, we wrote a spectral gating noise reduction algorithm [108]. The algorithm is inspired by the noise reduction algorithm used in the Audacity(R) sound editing software [109].

Given a waveform of audio with both signal and background noise ($S_n$), and a sample audio clip from the same or a similar waveform with only background noise ($N$). An outline of the algorithm is as follows:

1. Compute the short-time Fourier transform over $N$ ($spec_n$).

2. Compute the mean and standard deviation of $spec_n$ for each frequency component over time.

3. Compute the short-time Fourier transform over $S_n$ ($spec_s$).

4. For each frequency component, compute a threshold noise level based upon the mean and standard deviation of $spec_n$

5. Generate a mask over $spec_s$ based upon the power of $spec_s$ and the thresholds determined from ($spec_n$)

6. Smooth the mask over frequency and time.

7. Apply the mask to $spec_s$ to remove noise.

8. Compute the inverse short-time Fourier transform over $spec_s$ to generate a denoised time-domain signal.

We made a Python package of this algorithm called noisereduce available on GitHub [108]. In addition to the spectral gating noise reduction algorithm, segmentation was performed by a dynamic thresholding algorithm, which is described in the Segmentation section. We also show the fidelity of UMAP projections over different levels of noise in S6 Fig, where we observe that UMAP is robust to relatively high noise in comparison to spectrograms.

## Segmentation

Many datasets were made available with vocalizations already segmented either manually or algorithmically into units. When datasets were pre-segmented, we used the segment boundaries defined by the dataset authors. For all other datasets, we used a segmentation algorithm we call dynamic threshold segmentation (Fig 19A). The goal of the algorithm is to segment vocalization waveforms into discrete elements (e.g. syllables) that are defined as regions of continuous vocalization surrounded by silent pauses. Because vocal data often sits atop background noise, the definition for silence versus vocal behavior was set as some threshold in the vocal envelope of the waveform. The purpose of the dynamic thresholding algorithm is to set that noise threshold dynamically based upon assumptions about the underlying signal, such as the expected length of a syllable or a period of silence. The algorithm first generates a spectrogram, thresholding power in the spectrogram below a set level to zero. It then generates a vocal envelope from the power of the spectrogram, which is the maximum power over the frequency components times the square root of the average power over the frequency components for each time bin over the spectrogram:

$$\mu_s(t) = \frac{1}{n}\sum_f S(t,f) \tag{1}$$

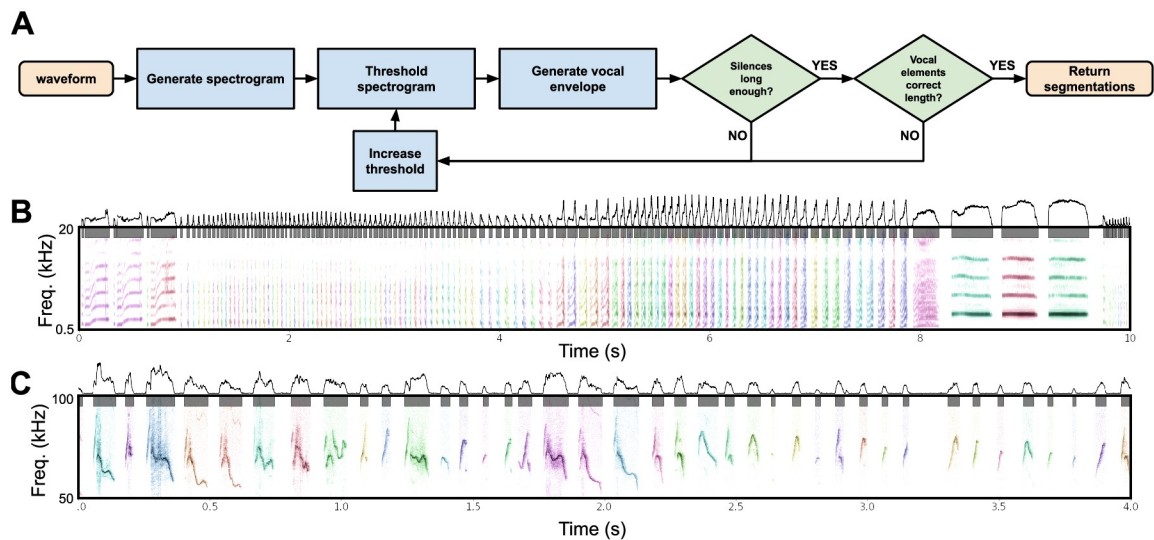

**Fig 19. Segmentation algorithm.** (A) The dynamic threshold segmentation algorithm. The algorithm dynamically defines a noise threshold based upon the expected amount of silence in a clip of vocal behavior. Syllables are then returned as continuous vocal behavior separated by noise. (B) The segmentation method from (A) applied to canary syllables. (C) The segmentation method from (A) applied to mouse USVs.

$$E(t) = \sqrt{\mu_s(t)} \max_f S(t, f) \qquad (2)$$

Where $E$ is the envelope, $S$ is the spectrogram, $t$ is the time bin in the spectrogram, $f$ is the frequency bin in the spectrogram, and $n$ is the total number of frequency bins.

The lengths of each continuous period of putative silence and vocal behavior are then computed. If lengths of vocalizations and silences meet a set of thresholds (e.g. minimum length of silence and maximum length of continuous vocalization) the algorithm completes and returns the spectrogram and segment boundaries. If the expected thresholds are not met, the algorithm repeats, either until the waveform is determined to have too low of a signal to noise ratio and discarded, or until the conditions are met and the segment boundaries are returned. The output of the algorithm, color coded by segment boundaries, are shown for a sample of canary song in Fig 19B and a sample of mouse USVs in Fig 19C. The code for this algorithm is available on Github [110].

## Spectrogramming

Spectrograms are created by taking the absolute value of the one-sided short-time Fourier transformation of the Butterworth band-pass filtered waveform. Power is log-scaled and thresholded using the dynamic thresholding method described in the Segmentation section. Frequency ranges and scales are based upon the frequency ranges occupied by each dataset and species. Frequency is logarithmically scaled over a frequency range using a Mel filterbank (a filterbank logarithmically scaled to match human frequency perception). Mel frequency scaling was used as it has previously proven useful in extracting features of animal vocalizations [111], although in some cases linear frequency scaling can perform better in bioacoustics [86]. All of the spectrograms we computed had a total of 32 frequency bins, scaled across frequency ranges relevant to vocalizations in the species. None of these parameters were rigorously compared

across each of the datasets, although we would recommend such comparisons in more detailed analyses. Each of the transformations done to data (e.g. downsampling, Mel scaling) reduce the dimensionality and impose *a priori* assumptions on the data. Performing analyses on a complete or invertible representation of the sound pressure waveform would make fewer assumptions [28], but is more computationally costly.

To create a syllable spectrogram dataset (e.g. for projecting into Fig 8), syllables are segmented from the vocalization spectrogram. To pad each syllable spectrogram to the same time length size, syllable spectrograms are log-rescaled in time (i.e. resampled in time relative to the log-duration of the syllable) then zero-padded to the length of the longest log-rescaled syllable.

## Projections

Latent projections are either performed over discrete units (e.g. syllables) or as trajectories over continuously varying sequences. For discrete units, syllables are segmented from spectrograms of entire vocalizations, rescaled, and zero-padded to a uniform size (usually 32 frequency and 32 time components). These syllables are then projected into UMAP, where each time-frequency bin is treated as an independent dimension.

Trajectories in latent space are projected from rolling windows taken over a spectrogram of the entire vocal sequence (e.g. a bout of birdsong; Fig 20). The rolling window is a set length in milliseconds (e.g. 5ms) and each window is treated as a single point to be projected into latent space. The window then rolls one frame (in the spectrogram) at a time across the entire spectrogram, such that the number of samples in a bout trajectory is equal to the number of time-frames in the spectrogram. These time bins are then projected into UMAP latent space.

## Clusterability

**Hopkin's statistic.**   We used the Hopkin's statistic [112] as a measure of the clusterability of datasets in UMAP space. In our case, the Hopkin's statistic was preferable over other metrics for determining clusterability, such as the Silhouette score [42] because the Hopkin's statistic does not require labeled datasets or make any assumptions about what cluster a data point should belong to. The Hopkin's statistic is part of at least one birdsong analysis toolkit [97].

The Hopkin's statistic compares the distance between nearest neighbors in a dataset (e.g. syllables projected into UMAP), to the distance between points from a randomly sampled dataset and their nearest neighbors. The statistic computes clusterability based upon the assumption that if the real dataset is more clustered than the randomly sampled dataset, points will be closer together than in the randomly sampled dataset. The Hopkin's statistic is computed over a set $X$ of $n$ data points (e.g. latent projections of syllables of birdsong), where the set $X$ is compared with a baseline set $Y$ of $m$ data points sampled from either a uniform or normal distribution. We chose to sample $Y$ from a uniform distribution over the convex subspace of $X$. The Hopkin's metric is then computed as:

$$\text{Hopkins statistic} = \frac{\sum_{i=1}^{m} w_i^d}{\sum_{i=1}^{m} u_i^d + \sum_{i=1}^{m} w_i^d} \tag{3}$$

Where $u_i$ is the distance of $y_i \in Y$ from its nearest neighbor in $X$ and $w_i$ is the distance of $x_i \in X$ from its nearest neighbor in $X$. Thus if the real dataset is more clustered than the sampled dataset, the Hopkin's statistic will approach 0, and if the dataset is less clustered than the randomly sampled dataset, the Hopkin's statistic will sit near 0.5. Note that the Hopkin's statistic is also commonly computed with $\sum_{i=1}^{m} u_i^d$ in the numerator rather than $\sum_{i=1}^{m} w_i^d$, where Hopkin's statistics closer to 1 would be higher clusterability, and closer to 0.5 would be closer to

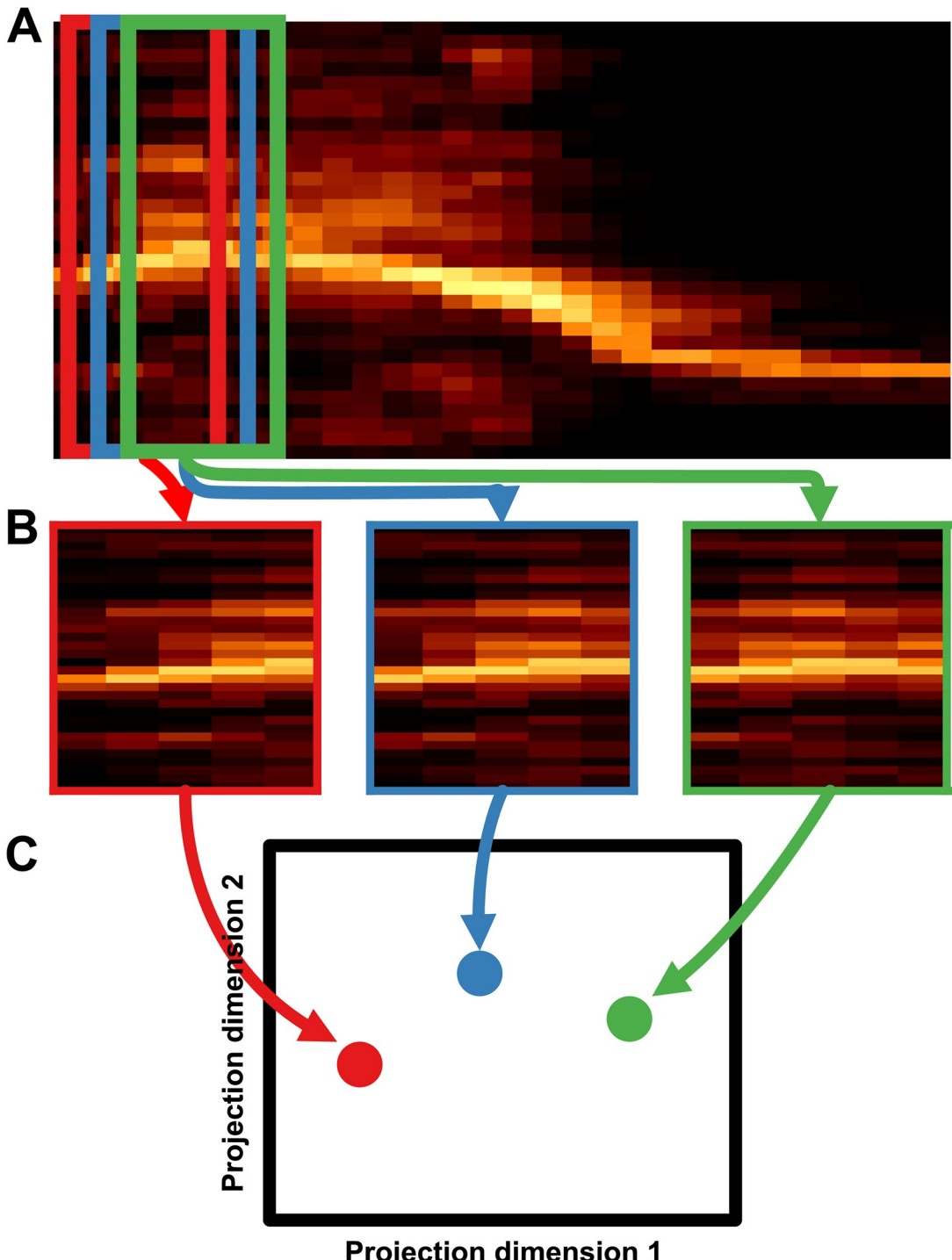

**Fig 20. Continuous projections from vocalizations.** (A) A spectrogram of each vocalization is computed. (B) Rolling windows are taken from each spectrogram at a set window length (here 5ms), and a step size of one time-frame of the short-time Fourier transform (STFT). (C) Windows are projected into latent space (e.g. UMAP or PCA).

chance. We chose the former method because the range of Hopkin's statistics across datasets were more easily visible when log transformed.

To compare the clusterability across songbirds and mammals, we used a likelihood ratio test between linear mixed-effects models predicting the Hopkin's statistic for each individual. Each model controlled for the number of vocalizations produced by individuals, and random variation in clusterability at the species level. In addition, we included only individuals that had recordings consisting of at least 200 vocalizations. The likelihood ratio test was performed between a model with, and without including class (i.e. songbird versus mammal) as a category.

**Silhouette score.** As opposed to the Hopkin's statistic, which measures the general clusterability of a projection without regard to cluster labels, the silhouette score measures the clusterability of datasets when cluster labels are known or have already been inferred [42]. In other words, the Hopkin's statistic measures how clusterable a projection is, and the silhouette score measures how well fit a clustering is to a projection.

The silhouette score, $S$ is computed as the mean of the silhouette coefficients for each data point. For each data point ($i$), the silhouette coefficient $s_i$ is the mean distance between the data point and all other data points in the same cluster ($a_i$), minus the distance to that points nearest neighbor belonging to a different cluster ($b_i$), divided by the maximum of $a_i$ and $b_i$, which can be written as:

$$s_i = \begin{cases} 1 - a_i/b_i, & \text{if } a_i < b_i \\ 0, & \text{if } a_i = b_i \\ b_i/a_i - 1, & \text{if } a_i > b_i \end{cases} \quad (4)$$

$$S = \frac{1}{n}\sum_{i=1}^{n} s_i \quad (5)$$

This value is therefore bounded between 1, when the average distance to other points within-cluster ($a_i$) is very large relative to the distance to the points nearest neighbor ($b_i$), and -1 when the average distance to points within-cluster ($a_i$) is very small relative to the distance to the nearest neighbor ($b_i$). Silhouette scores were compared across projections using a Kruskal-Wallis H-test over silhouette coefficients. Silhouette scores were compared to chance using a Kruskal-Wallis H-test over silhouette coefficients versus silhouette coefficients where labels are randomly permuted.

## Clustering vocalizations

**HDBSCAN.** HDBSCAN clustering was performed on PCA and UMAP projections of Cassin's vireo syllables and Bengalese finch syllables (Table 1), as well as UMAP projections of swamp sparrow syllables (Fig 11). Each clustering used the default parameterization of UMAP and HDBSCAN, setting the minimum cluster size at 1% of the number of syllables/notes in the dataset.

**K-means.** We used k-means as a comparison to hierarchical density-based labeling when clustering Bengalese finch and Cassin's vireo syllables (Table 1). K-means clustering partitions a set of data points into k-clusters by tiling data space with a set of cluster centroids, and clustering each data point with the nearest centroid. We set the number of clusters (k) to the ground truth number of clusters in each dataset, to make clustering more competitive with HDBSCAN. We used the k-means implementation in Scikit-learn [67] to fit the models.

**Gaussian Mixture model.** We clustered the known feature space of swamp sparrow song using a Gaussian Mixture Model (GMM). GMMs assume that data are generated from a mixture of finite Gaussian distributions. Our GMM fit the parameters of the distribution to the data using expectation-maximization. In both the Conneaut Marsh, PA and Hudson Valley, NY swamp sparrow datasets, we set the number of distributions to be equal to the numbers used in the same populations of swamp sparrows in [53]. As opposed to [53], we clustered only on the duration of the notes, and the start and end peak frequencies of the notes, without the mean peak frequency or vibrato amplitude. Still, these clusterings were similar to the clusterings presented in [53]. A direct comparison between our clustering using GMM and the clustering in [53] can be made by comparing Fig 11 and S4 Fig with Lachlan and Nowicki [53] S2A and S2B Fig. We used the GMM implementation in Scikit-learn [67] to fit the models.

## Comparing algorithmic and hand-transcriptions

Several different metrics can be used to measure the overlap between two separate labeling schemes. We used three metrics that capture different aspects of similarity to compare hand labeling to algorithmic clustering methods ([67, 68]; Table 1). Homogeneity measures whether all clusters fall into the same hand-labeled class in the labeled dataset.

$$\text{homogeneity}(\text{clusters, classes}) = 1 - \frac{H(\text{classes|clusters})}{H(\text{classes})} \tag{6}$$

Where $H(\text{classes|clusters})$ is the conditional entropy of the ground truth classes given the cluster labels, and $H(\text{classes}$ is the entropy of the classes.

Completeness measures the extent to which members belonging to the same hand-labeled class fall into the same cluster:

$$\text{completeness}(\text{clusters, classes}) = 1 - \frac{H(\text{clusters|classes})}{H(\text{clusters})} \tag{7}$$

V-measure is the harmonic mean between homogeneity and completeness.

$$\text{V-Measure} = 2 * \frac{\text{homogeneity} \cdot \text{completeness}}{\text{homogeneity} + \text{completeness}} \tag{8}$$

V-measure is also equivalent to the normalized mutual information between distributions [67]. In the swamp sparrow datasets, we compared the probability of overlap in clustering of labels (e.g. HDBSCAN and GMM) to chance by comparing V-measure of the true overlap to the bootstrapped V-measure permuting the clusterings (10,000 times).

## Hidden Markov Models (HMMs)

We used HMMs as a basis for comparing hand labels versus UMAP/HDBSCAN clustering in representing sequential organization in Bengalese finch song. Specifically, we treated hand labels as ground truth "visible" states in a discrete emission HMM, and generated several HMMs with different hidden states: hand labels, HDBSCAN labels, and hidden states learned using the Baum-Welch algorithm. HMMs were generated using the Python package Pomegranate [113]. Each model was compared on the basis of the log-likelihood of the data given the model. This log-likelihood score is treated as equal to the likelihood of the model given the data, and is also used as the basis computing AIC [71].

### Ethics statement

Procedures and methods comply with all relevant ethical regulations for animal testing and research and were carried out in accordance with the guidelines of the Institutional Animal Care and Use Committee at the University of California, San Diego (S05383).

## Supporting information

**S1 Fig. UMAP projections Cassin's vireo syllables with syllable features overlaid generated from the BioSound [24] python package.** (A) More information regarding each feature can be found in S2 Table and Elie et al. [24, 28].
(TIF)

**S2 Fig. Example vocal elements from each of the species used in this paper.**
(TIF)

**S3 Fig. Latent projections of vowels.** Each plot shows a different set of vowels grouped by phonetic features. The average spectrogram for each vowel is shown to the right of each plot.
(TIF)

**S4 Fig. Comparing latent and known features in swamp sparrow song.** (A) A scatterplot of the start and end peak frequencies of the notes produced by birds recorded in Hudson Valley, NY. The left panel shows notes colored by the position of each note in the syllable (red = first, blue = second, green = third). The center panel shows the sample scatterplot colored by a Gaussian Mixture Model labels (fit to the start and end peak frequencies and the note duration). The right panel shows the scatterplot colored by HDBSCAN labels over a UMAP projection of the spectrograms of notes. (B) The same notes, plotting the change in peak frequency over the note against the note's duration. (C) The same notes plotted as a UMAP projection over note-spectrograms. (D) The features from (A) and (B) projected together into a 2D UMAP space.
(TIF)

**S5 Fig. Comparison of Hidden Markov Model performance using different latent states.** Projections are shown for a single example bird from the Nicholson dataset [63]. UMAP projections are labeled by three labeling schemes: (A) Hand labels, (B) HDBSCAN labels on UMAP, and (C) Trained Hidden Markov Model (HMM) labels.
(TIF)

**S6 Fig. Silhouette score of UMAP projections with different levels background noise added to spectrogram.** White noise is added to the spectrogram to modulate signal to noise ratio (SNR). The different projections (2-dimensional PCA, 50-dimensional PCA, 2-dimensional UMAP) and the spectrogram are compared on the basis of silhouette score for the labels of each Cassin's vireo syllable. The left panel shows the silhouette score, and the right panel shows the silhouette score scaled between 0 and 1 to more easily compare change as a function of SNR.
(TIF)

**S1 Table. Overview of the species and datasets used in this paper.**
(PDF)

**S2 Table. BioSound features used in feature statistics analysis.** For more information see Elie et al. [24, 28].
(PDF)

## Acknowledgments

We would like to thank Kyle McDonald and his colleagues for motivating some of our visualization techniques with their work on humpback whale song [114].

## Author Contributions

**Conceptualization:** Tim Sainburg, Marvin Thielk, Timothy Q. Gentner.

**Data curation:** Tim Sainburg.

**Formal analysis:** Tim Sainburg.

**Funding acquisition:** Tim Sainburg, Timothy Q. Gentner.

**Investigation:** Tim Sainburg, Timothy Q. Gentner.

**Methodology:** Tim Sainburg, Marvin Thielk, Timothy Q. Gentner.

**Project administration:** Timothy Q. Gentner.

**Resources:** Tim Sainburg, Timothy Q. Gentner.

**Software:** Tim Sainburg, Timothy Q. Gentner.

**Supervision:** Timothy Q. Gentner.

**Validation:** Tim Sainburg, Marvin Thielk, Timothy Q. Gentner.

**Visualization:** Tim Sainburg.

**Writing – original draft:** Tim Sainburg.

**Writing – review & editing:** Tim Sainburg, Timothy Q. Gentner.

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
