## [Decision Letter · Decision Letter 0]

19 Feb 2020

Dear Mr. Sainburg,

Thank you very much for submitting your manuscript "Latent space visualization, characterization, and generation of diverse vocal communication signals" for consideration at PLOS Computational Biology.

As with all papers reviewed by the journal, your manuscript was reviewed by members of the editorial board and by several independent reviewers. In light of the reviews (below this email), we would like to invite the resubmission of a significantly-revised version that takes into account the reviewers' comments.

First I am very sorry that the review process took this long! Your paper came during the winter break and once we found reviewers they also took a long time to finish their review. As you will see they agree that the paper and the UMAP approach you use is useful for analyzing sounds. All reviewers raised some questions that are mostly about clarifications of the methods and that I believe you will be able to address easily (more details and a more "even level" presentation of all the methods) . There are also some questioning on the limitations of time-frequency representations (spectrograms) that are more open-ended questions but that do require attention. Maybe more significantly, your paper reads more like a very detailed tutorial for UMAP than a particular result. I believe it will be very useful for anyone who will be analyzing/categorizing natural sounds.

We cannot make any decision about publication until we have seen the revised manuscript and your response to the reviewers' comments. Your revised manuscript is also likely to be sent to reviewers for further evaluation.

Sincerely,

Frédéric E. Theunissen

Associate Editor

PLOS Computational Biology

Samuel Gershman

Deputy Editor

PLOS Computational Biology

Reviewer's Responses to Questions

**Comments to the Authors:**

Reviewer #1: Review uploaded as an attachment

Reviewer #2: Uploaded as attachment (12000 chars)

Reviewer #3: Review of Sainburg, Thielk and Gentner

This paper presents an overview of applications of deep latent representation learning to the study of vocal communication. The paper consists of three main parts. The first uses a recently introduced nonlinear dimensionality reduction tool (UMAP) to visualize structure in a large number of animal vocalization data sets. The second part reviews neural network methods that allow generation of samples from latent representations. The third part illustrates the application of one such method to behavioral and neurophysiological experiments in the starling.

This is an unusual paper, in that it is sort of mid-way between a review article and a research article. There are plenty of original results, but they are often impressionistic, given by visualizations that the reader is invited to inspect. In other cases (in the third part of the article), the results are somewhat anecdotal (and it is clear that a more thorough set of related results is being prepared for a dedicated research article). The paper feels more like an extended lecture than a typical modern-day scientific publication.

That said, I liked it. I found the approach refreshing and I liked the diversity, and found plenty of food for thought. The clusterability result of Figure 1 was cool, as was the comparative lack of clustering for speech (Figure 4). I suspect many people will enjoy reading this.

But there are some costs to the approach – for instance, the treatment of deep generative models is pretty brief, often at the expense of clarity. I think one could carve off part one of the paper and have enough for a substantial contribution. But I also think a paper with the current structure is viable (though it will need to be longer in places in my opinion). Because this is not a standard scientific paper with a question, approach, results and conclusions, my comments are mainly geared towards improving the presentation (more along the lines of what I would give for a review article).

Major comments:

1. The paper is very well-written. It feels like a lot of care was taken in the writing, and it shows. The introduction is excellent – clear and accessible.

2. The paper leans heavily on UMAP, and could benefit from a deeper description of it, either at the end of 2.1 or the start of 3.1. As it stands I think someone who doesn’t know about such methods will have a hard time contextualizing the results. I would like to see some equations and theoretical motivation.

In general, there is often a mismatch between the expertise needed to understand different parts of the paper. The paper seems aimed at the animal communication community, and so should have enough explanation throughout to be intelligible to someone from that community.

3. Similarly, there is heavy reliance on HDBSCAN, but almost no information about it. Most readers will want/need more.

4. Section 3.3 is similarly too brief. There are many places where an additional phrase or sentence is needed to explain the underlying idea. Here are a few places that I noticed:

-line 49 – the regularization loss and its relation to the latent distribution

-line 439 – the reconstruction loss in latent space

-it would be helpful here to refer to the variables in the figure, which I don’t believe are mentioned

-line 451 – adversarial loss function

-lines 451-452 – “train data…” – unclear – what problem does this solve?

-line 458 – costly in what sense?

5. There are a lot of figures, and they are frequently referred to out of order (and some are not referred to at all). Most journals require that figures, and individual figure panels, be referred to in order, and for good reason – it is much easier for the reader to be able to move through the text and figures in a coordinated way. This paper had pretty disjointed figure references.

6. The article relies heavily on the spectrogram, but does not discuss the potential issues in mapping from spectrograms to audio. Griffin and Lim often produces unavoidable artifacts (because some spectrograms do not correspond to a physically realizable sound signal). I would expect this to be a limitation of what the authors are doing in the last section. We should be told what they sound like, and also whether the synthesized signals produce latent representations that are close to those intended (which might in principle not be the case – the spectrogram of the generated audio signal is probably not going to be identical to the sampled spectrogram). There are in principle ways that this could be handled better, e.g. by optimizing the waveform rather than the spectrogram to produce the desired latent representation. Worth some discussion.

7. Related to point 6, the last section made me wonder whether the birds would produce similar behavior if trained only on the original exemplars. How much of the behavior is due to training on the synthetic morphs? If the synthetic sounds have artifacts, one might not expect great generalization from the original sounds to the interpolated sounds. But if similar results were obtained after training only on the originals, that would be impressive.

Minor comments:

Line 104: “control control” – typo

Line 169: “differentiable” seems like a poor choice in a paper that uses a lot of gradient descent. “discriminable” might be preferable.

Figure 3: caption is in wrong order

Section 3.1.3 – why is HDBSCAN not run on raw spectrograms?

Lines 262-264: can this effect be quantified in some way, rather than left for subjective evaluation?

Line 292: “Projection section” – where is this?

Figure 8 caption: The bits on panels M and N were unclear and hard to follow.

Line 356 – missing parenthetical

Line 473 – please give the motivation for the J-diagrams

Line 513 – “recording from extracellularly” – typo

Figure 15E is not mentioned in the text that I could see. What should we conclude from the comparison of 15E and 15F? And why are we not shown the similarity in the latent space in addition to the spectrogram? I might expect the similarity of the spectrograms to not be particularly well behaved, so maybe 15E is in fact showing similarity in the latent space?

I was surprised not to see mention of texture synthesis approaches, which are the one prior case I know of where realistic-sounding natural sounds (including animal choruses) were synthesized from rich latent representations. Some of the same things proposed here were successfully pulled off in that domain (e.g. experiments with morphs – see McWalter and McDermott 2018). Seems worth mentioning somewhere.

I would think that an obstacle to scaling up the proposed approach is the need for relatively clean audio recordings. This seems worth commenting on.

Figure 16 caption: “dynamically a noise threshold” – typo

Line 635 – why is there bandpass filtering? What are the parameters?

Lines 637-638: confusing. What is a “Mel filter”, and how does it produce a logarithmic scaling?

Lines 644-651: unclear – equations would help. What does “sampled as a single point” mean? What was the “set length in milliseconds”?

**Have all data underlying the figures and results presented in the manuscript been provided?**

Reviewer #1: Yes

Reviewer #2: Yes

Reviewer #3: Yes

PLOS authors have the option to publish the peer review history of their article (what does this mean?). If published, this will include your full peer review and any attached files.

Reviewer #1: Yes: Julie E Elie

Reviewer #2: No

Reviewer #3: No
---

## [Decision Letter · Decision Letter 1]

19 Jul 2020

Dear Mr. Sainburg,

Thank you very much for submitting your manuscript "Finding, visualizing, and quantifying latent structure across diverse animal vocal repertoires" for consideration at PLOS Computational Biology. As with all papers reviewed by the journal, your manuscript was reviewed by members of the editorial board and by several independent reviewers. The reviewers appreciated the attention to an important topic. Based on the reviews, we are likely to accept this manuscript for publication, providing that you modify the manuscript according to the review recommendations.

I am very sorry that this re-review took this long. Your referees were all very busy and it was a difficult time for them as you can imagine. Your paper was well received and is greatly improved. As you will see two of the reviewers are happy to see it on print as is. The second reviewer have a few minor points that I believe that you will be able to address rapidly by rewording. Point 3 should be addressed so that the readers don't over interpret or under interpret your results.

Looking forward to that final version ;-)

F.

Sincerely,

Frédéric E. Theunissen

Associate Editor

PLOS Computational Biology

Samuel Gershman

Deputy Editor

PLOS Computational Biology

[LINK]

Dear Tim,

First, I am very sorry that this re-review took this long. Your referees were all very busy and it was a difficult time for them as you can imagine. Your paper was well received and is greatly improved. As you will see two of the reviewers are happy to see it on print as is. The second reviewer have a few minor points that I believe that you will be able to address rapidly by rewording. Point 3 should be addressed so that the readers don't over interpret or under interpret your results.

Looking forward to that final version ;-)

F.

Reviewer's Responses to Questions

**Comments to the Authors:**

Reviewer #1: The authors have done a very good job in the revision of this manuscript that is now more focused around a same objective/story: the potential of latent projections for the study of animal communication. This paper is didactic and I like in particular the effort made with the new figures and sections of the paper (introduction on UMAP, discussion/limitations of the approach).

This will definitely be a very useful contribution to the field!

I spotted some typos that the authors will want to correct:

Line 68: suppress comma : Kershenbaum et. al., [1] -> Kershenbaum et. al. [1]

Figure 1: in the legend correct: UMA,P -> UMAP

Figure 3: revise legend: “(E) UMAP of spectrograms where the color for each syllables where color is the syllable’s average fundamental frequency (F) The same as (E) where pitch saliency of each syllable, which corresponds to the relative size of the first auto-correlation peak represents color.”

Line 305: Fig5C -> reference to wrong figure

Line 335-343: Text paced twice

Line 360-380: Check consistency for spelling “v-measure” or “V-Measure”

Line 455: Fig13 D,J -> Fig 13C,J?

Figure 14 : Error in legend reference to subplot C and D

Figure 15: “The same plots as in (A),” -> “the same data as in A-C”?

Figure 15: “The same plot as (D-E)” -> “the same plotS as (D-F)”

Figure 15: Can you add the time color scale used in A-C at the bottom of spectrograms in M and N so we know what color represent the beginning/end of the bout?

Figure 20: Make sure you define “STFT” (not everyone might guess this is Shot Term Fourrier Transform)

Figure 16: “corresponds is” -> “corresponds to”

Discussion about how you choose 200ms as the optimal size of the rolling window for Starlings?

Line 557: to categories -> “to categorize”

Line 564-565: Specify what the distance metric is: Euclidean distance between latent projection vectors?

Figure 17E: Specify the size of the scale bar: 100ms?

Line 596: “more smooth” -> “smoother”

Line 737: missing word: In this ?

Line 938: You probably want to remove that sentence: "The behavioral and neural data are part of a larger project and will be released alongside that manuscript."

Reviewer #2: Review uploaded as an attachment

Reviewer #3: Re-review of Sainburg, Thielk, Gentner

The revision of this paper is improved. I recommend publication after the authors address my comments as they see fit (all minor). I don’t need to see the paper again.

The most substantial revision the authors should consider is to lengthen and improve the description of U-MAP at the start. I fear the description they give is still not sufficient to give someone who is not already familiar with such methods a sense for how/why it works. It would help to spell out what it means to find an embedding that preserves the structure of the graph.

Another important point is that the authors should emphasize at the outset that U-MAP does not use labels. This is not made explicit on page 6. It is stated later on, but should be prominently mentioned earlier.

I love the analysis of co-articulation. Very cool.

Other small things I spotted:

163: “problematically” – wrong word

228: order of figure numbers is wrong

282-284: specify which version is shown in the figure (UMAP on spectrograms or power spectra)

376-377: what is the difference between “sub-clusters of the hand labels” and “hand labels that are split into multiple categories”? unclear to me.

404: “similar level of overlap with position of notes” – confusing

423: “element” – typo

557: “categories” – typo

Fig 17F is blank

650: “behavioral: - typo

734-735: “Determining…” – something got garbled

737: “In this ,” – missing word

788: “deficit” – wrong word

**Have all data underlying the figures and results presented in the manuscript been provided?**

Reviewer #1: Yes

Reviewer #2: Yes

Reviewer #3: Yes

PLOS authors have the option to publish the peer review history of their article (what does this mean?). If published, this will include your full peer review and any attached files.

Reviewer #1: **Yes: **Julie E Elie

Reviewer #2: No

Reviewer #3: No
---

## [Editor Report · Decision Letter 2]

8 Aug 2020

Dear Mr. Sainburg,

We are pleased to inform you that your manuscript 'Finding, visualizing, and quantifying latent structure across diverse animal vocal repertoires' has been provisionally accepted for publication in PLOS Computational Biology.

Best regards,

Frédéric E. Theunissen

Associate Editor

PLOS Computational Biology

Samuel Gershman

Deputy Editor

PLOS Computational Biology

Dear Tim,

Looking forward to seing your paper in the journal so that it can be cited and distributed.

Best,

Frederic.

---

## [Editor Report · Acceptance letter]

2 Oct 2020

PCOMPBIOL-D-19-02154R2 

Finding, visualizing, and quantifying latent structure across diverse animal vocal repertoires

Dear Dr Sainburg,

I am pleased to inform you that your manuscript has been formally accepted for publication in PLOS Computational Biology. Your manuscript is now with our production department and you will be notified of the publication date in due course.

With kind regards,

Laura Mallard
